# Isoform-specific knockdown of long and intermediate prolactin receptors interferes with evolution of B-cell neoplasms

Adeleh Taghi Khani [1], Anil Kumar[1], Ashly Sanchez Ortiz[1], Kelly C. Radecki[2], Soraya Aramburo[1], Sung June Lee[1], Zunsong Hu [1], Behzad Damirchi[1], Mary Y. Lorenson[2], Xiwei Wu[3], Zhaohui Gu [1,4], William Stohl [5], Ignacio Sanz [6], Eric Meffre[7], Markus Müschen [8], Stephen J. Forman [9,10,11], Jean L. Koff [12], Ameae M. Walker [2]✉ & Srividya Swaminathan [1,11]✉

Prolactin (PRL) is elevated in B-cell-mediated lymphoproliferative diseases and promotes B-cell survival. Whether PRL or PRL receptors drive the evolution of B-cell malignancies is unknown. We measure changes in B cells after knocking down the pro-proliferative, anti-apoptotic long isoform of the PRL receptor (LFPRLR) in vivo in systemic lupus erythematosus (SLE)- and B-cell lymphoma-prone mouse models, and the long plus intermediate isoforms (LF/IFPRLR) in human B-cell malignancies. To knockdown LF/IFPRLRs without suppressing expression of the counteractive short PRLR isoforms (SFPRLRs), we employ splice-modulating DNA oligomers. In SLE-prone mice, LFPRLR knockdown reduces numbers and proliferation of pathogenic B-cell subsets and lowers the risk of B-cell transformation by downregulating expression of activation-induced cytidine deaminase. LFPRLR knockdown in lymphoma-prone mice reduces B-cell numbers and their expression of BCL2 and TCL1. In overt human B-cell malignancies, LF/IFPRLR knockdown reduces B-cell viability and their MYC and BCL2 expression. Unlike normal B cells, human B-cell malignancies secrete auto-crine PRL and often express no SFPRLRs. Neutralization of secreted PRL reduces the viability of B-cell malignancies. Knockdown of LF/IFPRLR reduces the growth of human B-cell malignancies in vitro and in vivo. Thus, LF/IFPRLR knockdown is a highly specific approach to block the evolution of B-cell neoplasms.

[1] Department of Systems Biology, Beckman Research Institute of City of Hope, Monrovia, CA 91016, USA. [2] Division of Biomedical Sciences, School of Medicine, University of California, Riverside, Riverside, CA 92521, USA. [3] Department of Molecular and Cellular Biology, City of Hope National Medical Center, Duarte, CA 91010, USA. [4] Department of Computational and Quantitative Medicine, Beckman Research Institute of City of Hope, Duarte, CA 91010, USA. [5] Division of Rheumatology, Department of Medicine, Keck School of Medicine of the University of Southern California, Los Angeles, CA 90033, USA. [6] Department of Medicine, Division of Rheumatology, Lowance Center for Human Immunology, Emory University, Atlanta, GA 30322, USA. [7] Division of Immunology and Rheumatology, Stanford University School of Medicine, Stanford, CA 94305, USA. [8] Center of Molecular and Cellular Oncology, Yale School of Medicine, 300 George Street, 06520 New Haven, CT, USA. [9] Department of Hematology & Hematopoietic Cell Transplantation, City of Hope National Medical Center, Duarte, CA 91010, USA. [10] Department of Immuno-Oncology, Beckman Research Institute of City of Hope, Duarte, CA 91010, USA. [11] Department of Pediatrics, Beckman Research Institute of City of Hope, Duarte, CA 91010, USA. [12] Department of Hematology and Medical Oncology, Emory University School of Medicine, Atlanta, GA 30322, USA. ✉email: ameae.walker@ucr.edu; sswaminathan@coh.org

B-cell malignancies, such as diffuse large B-cell lymphoma (DLBCL) and acute lymphoblastic leukemia (ALL), have widely divergent outcomes[1,2]. To identify early interventions for high-risk B-cell malignancies, we are delineating the mechanisms underlying the evolution of these neoplasms, which include malignancy initiation, overt neoplasm establishment, and maintenance of overt malignancy.

Because the initiation of B-cell malignancies is >3-fold more frequent in patients with the B cell-mediated autoimmune disease, systemic lupus erythematosus (SLE), than the healthy population[3-5], SLE-prone mouse models represent robust biological systems to identify signaling pathways that elevate the risk of B-cell transformation[6]. Similarly, because B-cell cancers often clonally evolve from pre-malignant and indolent B-cell clones[7-9], transgenic mice prone to developing B-cell neoplasms are robust models for investigating the mechanisms that promote the establishment of overt B-cell malignancies. For analysis of sustenance of overt B-cell malignancies, we have used cell lines and samples derived from patients with these neoplasms.

Prolactin (PRL), a sex hormone, classically recognized for its crucial role in lactation but produced by both sexes in response to acute and chronic stress[10], can drive pathological processes including tumorigenesis[11,12]. We postulated that PRL promotes the evolution of B-cell malignancies because it (1) promotes the survival of autoreactive B cells[13] and is associated with the exacerbation of SLE[14,15], a disease that increases the risk of B-cell malignancy initiation[3-5], (2) enhances the survival of normal and pathogenic mouse B cells and induces their expression of the proto-oncogenes c-MYC (cellular Myelocytomatosis) and BCL2 (B-cell lymphoma 2) in vitro[16-19], and (3) complexes with the immunoglobulin G (IgG) heavy chain to promote proliferation of malignant B cells in chronic lymphocytic leukemia (CLL), a malignancy associated with autoimmune manifestations[20].

Murine and human PRL receptors (PRLRs) are cytokine receptors with long (LF) and short (SF) isoforms, generated by alternative splicing[21,22]. Humans have an additional intermediate splice isoform (IF), at least in pathological tissues[23]. These different PRLRs have identical extracellular and transmembrane domains but differ in their intracellular domains such that they initiate different signals. A major signaling difference between the two categories of receptor is that the LFPRLR can activate the Janus kinase/signal transducer and activation of transcription (Jak/STAT) pathway, whereas the shorter cytoplasmic domains of the SFPRLRs cannot engage STAT[11]. Although the IF cannot engage STATs, it initiates STAT-independent pro-tumorigenic signaling upon dimerization with LF[23,24]. Overall, the increased expression of LF/IF relative to SFPRLRs on cells causes cell proliferation and survival, whereas increased expression of SF relative to LF/IF inhibits proliferation and induces apoptosis[23,25,26]. Thus, LF/IFPRLRs are antagonistic to SFPRLRs.

Whether PRL drives the evolution of overt B-cell malignancies via specific PRLR isoforms, is unknown. We hypothesized that, PRL enhances the risk of clonal evolution of autoimmune and pre-malignant B cells and sustains overt B-cell malignancies by signaling through the pro-proliferative and anti-apoptotic LF/IFPRLR. To test this, we measured the effects of systemically knocking down expression of the LFPRLR on immune cells: (1) in vivo in mouse models prone to SLE carrying non-malignant but aberrant B cells[27], (2) in vivo in mouse models prone to DLBCL having pre-malignant B cells[28], and (3) in vitro and in vivo in overt malignant human B-cell lines along with knockdown of IFPRLR.

For isoform-specific knockdown of LFPRLR in mice and LF/IFPRLR in human cells, we employed an innovative and tractable approach involving 25mer splice-modulating deoxyribonucleic acid morpholino oligomers (SMOs), termed here 'LFPRLR SMO',

that selectively prevent splicing to produce the LFPRLR in mice and LF and IF PRLRs in humans. More specifically, the SMOs prevent the inclusion of exon 10 during splicing of *PRLR* pre-messenger ribonucleic acid (pre-mRNA) that encodes an intracellular domain in the LFPRLR required for binding of STAT proteins. The sequence of LFPRLR SMO is different for mouse and human[29]. The SMOs are highly stable, non-immunogenic deoxyribonucleic acid (DNA) oligomers linked to an octaguanidine dendrimer that ensures effective whole-body uptake. They are administered subcutaneously and have been found to be efficacious and non-toxic in healthy mice and syngeneic and human xenograft models of breast cancer[29,30].

We found that signaling of PRL through the LF/IFPRLR raises the risk of B-cell malignancy initiation in SLE-prone mice, increases the survival of pre-malignant B cells in DLBCL-prone mice, and sustains the progression of established human B-cell malignancies. High expression of PRL in tumors associates with poor clinical prognosis in patients with B-cell malignancies. We identify blocking synthesis of the LF/IFPRLR using LFPRLR SMO as an isoform-specific and clinically attractive strategy that merits further investigation as a potential treatment for autoimmune and malignant B cell-mediated lymphoproliferative diseases.

## Results

**Knockdown of LFPRLR reduces splenic B-cell subsets in SLE-prone mice.** To investigate whether the PRL-LFPRLR axis raises the risk of initiation of B-cell malignancies, we compared SLE-prone *MRL-lpr* mice treated with either control SMO or LFPRLR SMO. Among SLE-prone models, we chose *MRL-lpr* mice because they accumulate genetic lesions indicative of B-cell transformation[6] and exhibit the spontaneous lymphoproliferation mutation *Fas^{lpr}*[27] that has been implicated in lymphoma pathogenesis[31,32]. Female *MRL-lpr* mice succumb to SLE between 17–22 weeks postnatally[27]. Therefore, we treated female mice with control SMO or LFPRLR SMO from the age of 6 weeks until 14 weeks when we quantified expression of molecular indicators known to raise the risk of B-cell malignancy development (Fig. 1a).

Because no commercially available antibodies could detect endogenous expression of PRLR isoforms in our models, we used qPCR to measure the absolute levels of each isoform. Of the mouse *Prlr* isoforms, LF, SF1, SF2, and SF3[22], only SF3 and LF were found to be expressed in splenic leukocytes of *MRL-lpr* mice. The ratio of LF *Prlr*: total *Prlr* mRNA, which indicates the level of pro-proliferative and/or anti-apoptotic PRLR signaling in cells, was significantly downregulated in splenic white blood cells (WBCs) of *MRL-lpr* mice treated with LFPRLR SMO (Fig. 1b). This confirmed LFPRLR knockdown in immune cells.

Because splenic T-, B-, and dendritic cells (DCs) drive SLE pathology[33], we examined changes in these WBCs after LFPRLR knockdown by flow cytometry (Supplementary Fig. 1). WBC, B-cell, plasmacytoid (pDC), and conventional DC (cDC) counts were significantly reduced after LFPRLR knockdown, whereas counts of T cells and T-cell subsets (CD4+, CD8+, and pathologic CD4−CD8− SLE T cells) remained unaltered (Fig. 1c–e, Supplementary Fig. 2).

B-cell pathology in SLE is largely driven by autoantibody-producing Blimp1+CD138+B220+ short-lived plasmablasts and Blimp1+CD138+B220− long-lived plasma cells (LLPCs)[34]. B cells mature into plasma cells with help from DCs[35-41]. Reductions in B cells and DCs after LFPRLR knockdown suggested that treatment with LFPRLR SMO may affect plasma cell subsets. As predicted, plasmablast and LLPC numbers were significantly reduced after LFPRLR knockdown (Fig. 1f, Supplementary Fig. 3a).

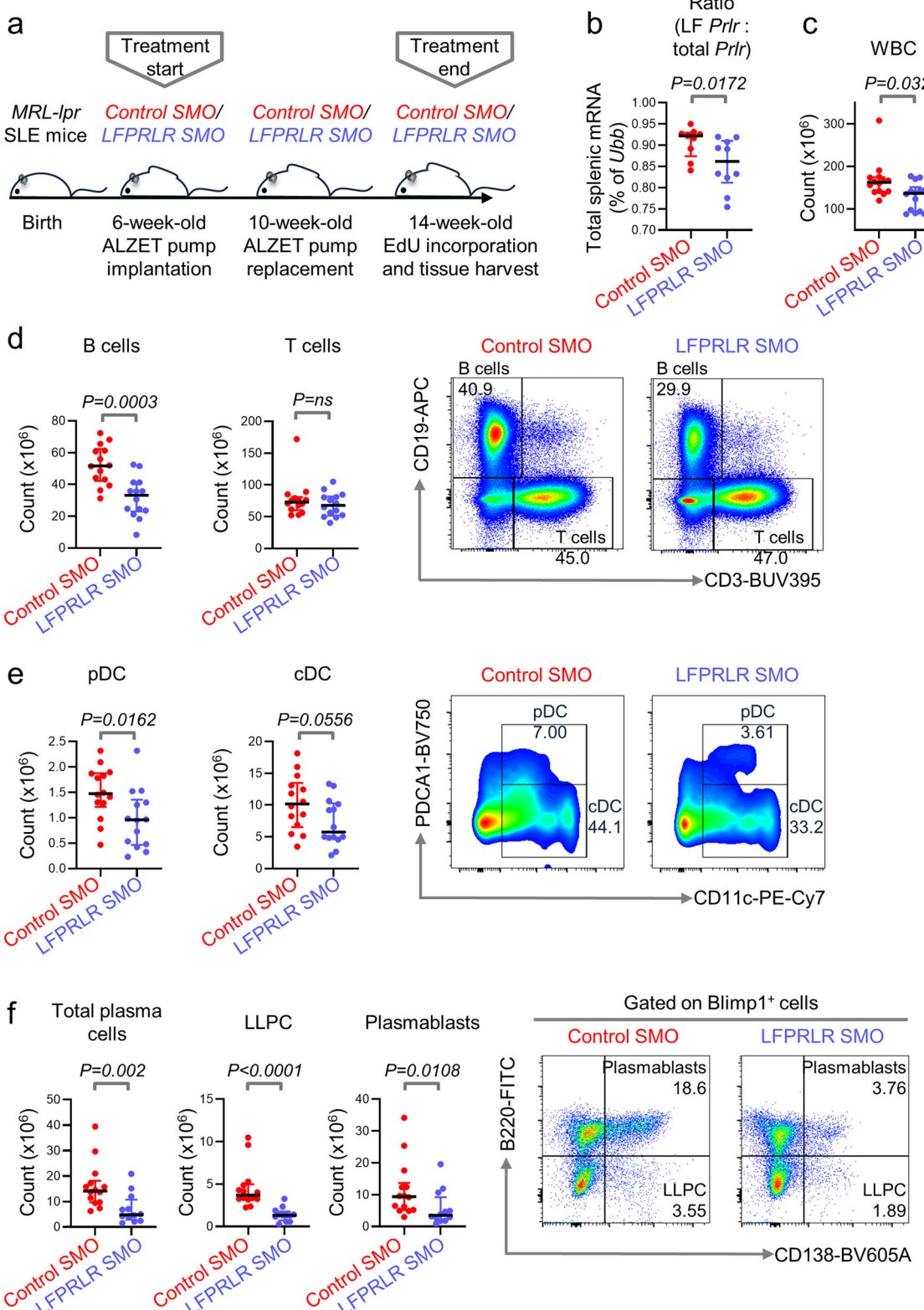

**Fig. 1 Knockdown of LFPRLR reduces splenic B-cell subsets in SLE-prone mice. a–c** Experimental design (**a**), verification of LFPRLR knockdown in total splenic WBCs by qPCR (**b**), and splenic WBC counts (**c**) in *MRL-lpr* mice treated with control SMO [$n = 9$ for (**b**) and $n = 14$ for (**c**), red] or LFPRLR SMO [$n = 10$ for (**b**) and $n = 14$ for (**c**), blue]. **d–f** Quantitation and representative flow cytometry plots of CD3−CD19+ B cells and CD19−CD3+ T cells (**d**); CD11c+PDCA1+ pDCs and CD11c+PDCA1− cDCs (**e**); Blimp1+CD138+ total plasma cells, Blimp1+CD138+B220+ plasmablasts, and Blimp1+CD138+B220− long lived plasma cells (**f**) in spleens of *MRL-lpr* mice treated with control SMO ($n = 14$, red) or LFPRLR SMO ($n = 12$, blue). Graphs show median ± interquartile range. Each dot in qPCR analyses of transcripts corresponds to the mean expression of that gene in one mouse calculated from 3 technical replicates. Ubiquitin B (*Ubb*) was used as the house-keeping gene in qPCR. Exact *p*-values were calculated using the Mann–Whitney *U* test. ns non-significant.

As SLE progresses, there is an increase in production of autoantibodies such as those against double-stranded (ds) DNA[42]. Furthermore, a recent study showed that PRL drives the production of anti-dsDNA autoantibodies in *MRL-lpr* mice[43]. Whether PRL promotes such autoantibody production by signaling specifically through the LFPRLR was not determined. Hence, we determined whether LFPRLR knockdown impacted the production of anti-dsDNA autoantibodies in serum of SLE-prone mice. Although the early stage of disease analyzed makes the *Mrl-lpr* model less than ideal for this measurement, levels of anti-dsDNA autoantibodies trended towards reduction after LFPRLR knockdown (Supplementary Fig. 4), which matches with the reduction in plasma cell subsets seen earlier. Our findings lead us to conclude that LFPRLR knockdown reduces numbers of pathologic splenic B-cell subsets in SLE-prone mice.

**Knockdown of LFPRLR in SLE-prone mice decreases factors associated with the risk of lymphoma initiation**. We examined B-cell turnover and composition of the mature B-cell repertoire in SLE-prone mice post LFPRLR knockdown. We confirmed that treatment with LFPRLR SMO reduces the *LF Prlr*: total *Prlr* ratio in magnetically sorted splenic B cells (Supplementary Fig. 5) by qPCR (Fig. 2a). We then measured B-cell proliferation and found that LFPRLR knockdown reduced the percentage of B cells in S phase while concomitantly increasing the proportion of G0/G1 B cells (Fig. 2b). LFPRLR knockdown did not increase the frequency of pre-apoptotic sub-G0/G1 B cells, concordant with *Fas* dysfunctionality in *MRL-lpr* mice that prevents the death of arrested B cells.

Because PRL induces the expression of MYC and BCL2 in lymphocytes[16–19], we examined the expression of these proto-oncogenes in splenic B cells after LFPRLR knockdown. We found no changes in *Myc*, significantly reduced *Bcl2* mRNA, but no changes in BCL2 protein in the time frame of treatment (Fig. 2c, d, Supplementary Fig. 3b). The anti-apoptotic protein BCL2 is known to promote B-cell survival but not proliferation[44]. Our findings showing unchanged expression of BCL2 protein and reduction in B-cell proliferation after LFPRLR knockdown in SLE-prone mice are consistent with this. However, the significant reduction in *Bcl2* mRNA suggests the possibility of longer-term downregulation of BCL2 protein by LFPRLR knockdown.

Next, we investigated whether LFPRLR induced indicators of evolution of autoimmune B cells into malignancies by comparing the splenic B-cell repertoire of SLE-prone mice treated with control SMO ($n = 6$) or LFPRLR SMO ($n = 6$) using immunoglobulin heavy chain (*IGH*) sequencing. A healthy B-cell repertoire has very few deleterious, potentially polyreactive B-cell clones with *IGH* complementary determining region 3 (CDR3) of lengths ≥20 amino acids (≥60 nucleotides)[45–48] and B cells carrying non-functional *IGH* rearrangements with an increased propensity towards malignant transformation[9]. Although the B-cell repertoire was normally distributed after LFPRLR knockdown, frequencies of potentially abnormal B cells with long CDR3s ranging from 35-42 amino acids, and those with non-functional *IGH* rearrangements were significantly reduced in LFPRLR SMO-treated SLE-prone mice (Fig. 2e–g).

B-cell malignancies evolve from hyperactivation of, and off-target genomic alterations induced by, the B-cell enzyme, activation induced cytidine deaminase (AID, *Aicda* gene)[9,49]. AID diversifies the B-cell repertoire through somatic hypermutation (SHM) and class switch recombination (CSR)[50]. We hypothesized that PRL-LFPRLR signaling may induce AID expression because (1) estradiol, which promotes *Aicda* transcription and accelerates malignant transformation of B-lymphocytes in *MRL-lpr* mice[51] could, in part, mediate this process via stimulation of PRL secretion, and (2) a reduction in both DCs that mediate CSR, and LLPC that result from class-switched B cells in LFPRLR SMO-treated SLE-prone mice suggested that PRL-LFPRLR signaling may regulate AID expression. As anticipated, expression of *Aicda* mRNA and AID protein was significantly reduced in splenic B cells after LFPRLR knockdown (Fig. 2h, Supplementary Fig. 3c). Hence, PRL signaling through LFPRLR primes B cells for acquisition of oncogenic mutations by promoting AID expression. Thus, the PRL-LFPRLR axis promotes the retention of deleterious B cells, thereby increasing the pool of B cells available for malignant transformation and malignancy initiation.

**Knockdown of LFPRLR impacts early B-lymphopoiesis in SLE-prone mice**. We determined the extent to which reduced splenic B cells after LFPRLR knockdown in SLE-prone mice resulted from changes in early B-lymphopoiesis. Although bone marrow B-cell counts were unchanged after LFPRLR knockdown, their expression of BCL2 was strikingly reduced (Fig. 3a, b). Because BCL2 is critical for survival of immature B cells[52], including autoreactive B cells[46,53,54], we conclude that the LFPRLR likely contributes to the inappropriate survival and retention of abnormal, early B cells in SLE-prone mice.

Overall, LFPRLR drives inappropriate early and mature B-lymphopoiesis in SLE-prone mice; the former mediated via induction of BCL2 expression and the latter by increasing DC numbers, which can in turn promote mature B-cell cycling, AID expression, and plasma cell generation. Consequently, deleterious B cells with non-productive *IGH* rearrangements and long CDR3s are retained in the repertoire, thereby raising the risk of lymphomagenesis (Fig. 3c).

**Knockdown of LFPRLR in DLBCL-prone mice suppresses factors that can drive overt B-cell lymphomagenesis**. To determine whether the PRL-LFPRLR axis elevates the risk of establishment of overt B-cell malignancies from pre-malignant B cells, we compared B-cell pathology in DLBCL-prone *CD79b-TCL1-tg* mice[28], treated with either control SMO or LFPRLR SMO. All mice hemizygous for the T-cell leukemia/lymphoma protein 1 (*TCL1*) transgene develop signs of increased risk of DLBCL beginning at 4 months of age. These include mild lymphocytosis and slight increases (<2-fold) in WBC counts, without changes in lymphoid organ size or structure, or changes in early or late B-cell lymphopoiesis. However, the majority (~95%) of mice become visibly ill with symptoms of overt DLBCL between 7–12 months of age[28]. Therefore, to examine the role of LFPRLR in raising the risk of development of overt lymphoma from pre-malignant B cells, we treated 8-week-old *TCL1-tg* mice with the SMOs until 16-weeks-old (Fig. 4a), which, in every mouse, is before the development of overt DLBCL. As in *MRL-lpr* SLE-prone mice, only SF3 and LF were found to be expressed in splenic leukocytes of *TCL1-tg* DLBCL-prone mice. We first confirmed LFPRLR knockdown in total splenic WBCs by qPCR (Fig. 4b). LFPRLR knockdown significantly reduced the numbers of splenic WBC in *TCL1-tg* DLBCL-prone mice (Fig. 4c). Among splenic WBC subsets, LFPRLR knockdown significantly reduced B-cell counts but not counts of total T cells, T-cell subsets, or DC subsets (Fig. 4d, Supplementary Fig. 6). LFPRLR knockdown also significantly reduced expression of the driver TCL1 oncoprotein[28] in splenic B cells in the DLBCL-prone mice (Fig. 4e, Supplementary Fig. 3d). Hence, abrogation of LFPRLR synthesis in 16-week-old *TCL1-tg*-mice specifically impacts

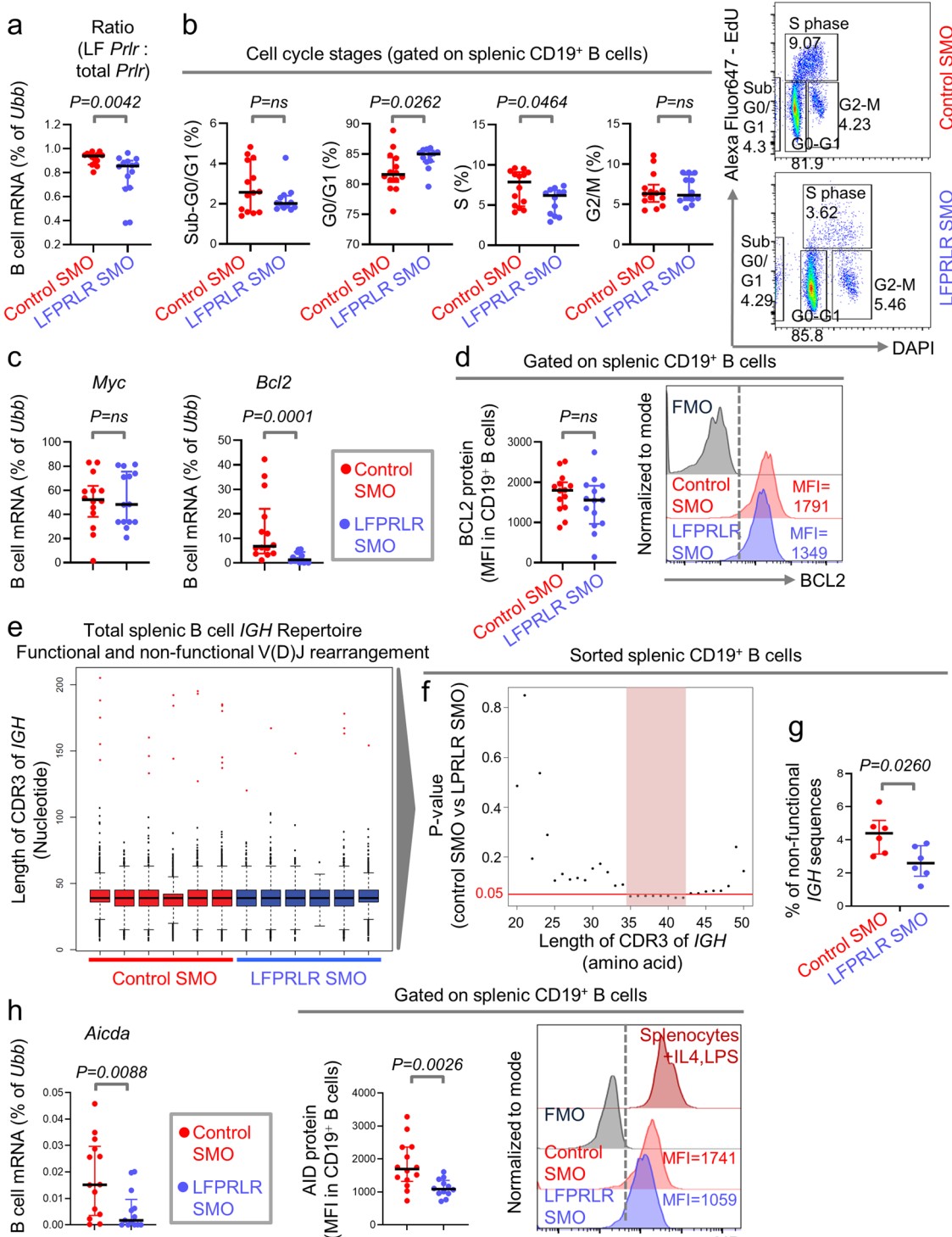

**Fig. 2 Knockdown of LFPRLR in SLE-prone mice decreases factors associated with the risk of lymphoma initiation. a–d** Verification of LFPRLR knockdown in isolated B cells by qPCR (**a**), percentages of B cells in sub G0/G1, G0/G1, S, and G2/M phases with representative flow cytometry plots (**b**), B-cell specific transcript levels of *Myc*, *Bcl2* by qPCR (**c**), and BCL2 protein by flow cytometry (**d**) in spleens of *MRL-lpr* mice treated with control SMO ($n = 14$, red) or LFPRLR SMO ($n = 14$, blue). **e–g** Next-generation sequencing of total *IGH* repertoire (GSE207186) to compare spectratype of splenic B cells (**e**), and frequencies of splenic B cells with *IGH* CDR3 ≥ 20aa (**f**), and frequencies of B cells with non-productive *IGH* rearrangements (**g**) in *MRL-lpr* SLE-prone mice treated with control ($n = 6$, red) or LFPRLR SMO ($n = 6$, blue). **h** *Aicda* mRNA and protein levels in splenic B cells and representative histograms for AID protein staining in control SMO-treated ($n = 14$, red) and LFPRLR SMO-treated ($n = 14$, blue) *MRL-lpr* mice by flow cytometry. Splenic WBCs from wildtype mice cultured with IL-4 and LPS were used as positive controls for AID staining. Each dot in qPCR analyses of transcripts corresponds to the mean expression of that gene in one mouse calculated from 3 technical replicates. *Ubb* was used as the house-keeping gene in qPCR. Graphs show median ± interquartile range. Exact p-values were calculated using the Mann–Whitney *U* test. ns non-significant, FMO fluorescence minus one control, MFI median fluorescence intensity.

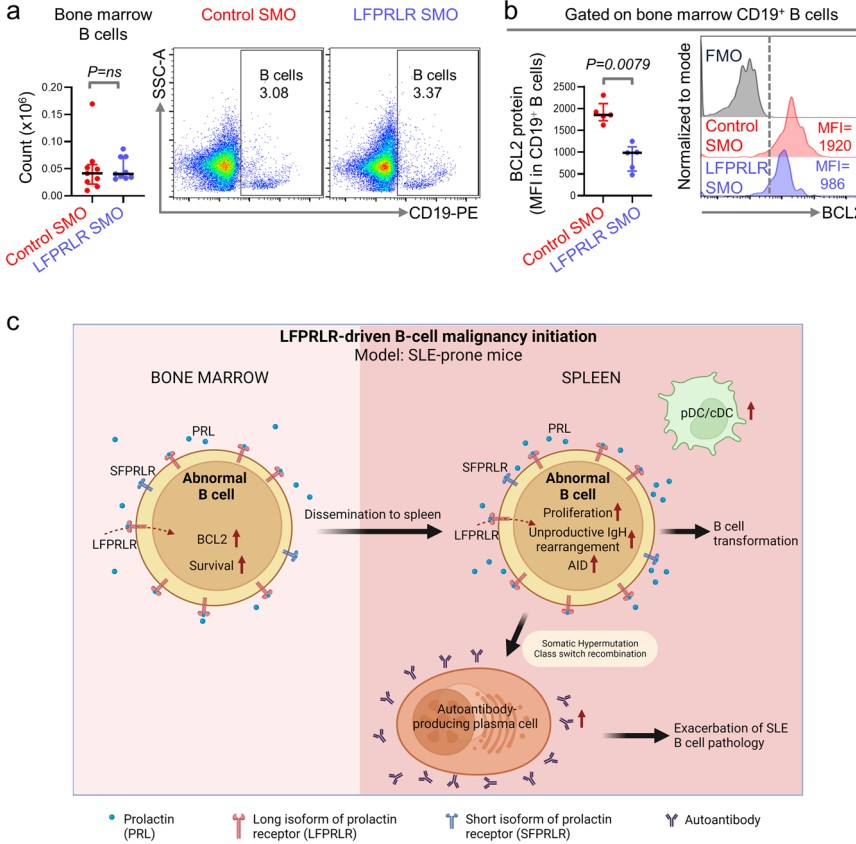

**Fig. 3 Knockdown of LFPRLR impacts early B-lymphopoiesis in SLE-prone mice. a** Quantitation and representative flow cytometry plots of bone marrow CD19⁺ B cells in *MRL-lpr* mice treated with control SMO (*n* = 9, red) or LFPRLR SMO (*n* = 9, blue). **b** Quantitation and representative histograms showing median fluorescence intensity (MFI) of BCL2 protein in bone marrow B cells of control SMO-treated (*n* = 5, red) and LFPRLR SMO-treated (*n* = 5, blue) *MRL-lpr* mice by flow cytometry. **c** A model depicting the mechanisms by which LFPRLR increases the number of abnormal splenic B cells available for potential malignant transformation or the concurrent worsening of autoimmune pathology in SLE-prone mice. Graphs show median ± interquartile range. Exact *p*-values were calculated using the Mann–Whitney *U* test. ns non-significant, FMO fluorescence minus one control. **c** was created with BioRender.com.

pre-malignant B cells and reduces the major indicators of B-cell lymphoma risk specific to this age group. These findings strengthened the rationale for further interrogating the causal mechanisms by which LFPRLR raises the risk for establishment of overt B-cell neoplasms.

B-cell cycling was unaffected after LFPRLR knockdown in *TCL1-tg* mice (Supplementary Fig. 7). To examine whether LFPRLR knockdown modulated expression of *Myc* and *Bcl2* mRNA, we initially used splenic WBCs because we could sort only a limited number of B cells even after pooling samples from ~3–4 mice per group. *Myc* expression was unchanged but that of *Bcl2* mRNA was significantly downregulated after LFPRLR knockdown (Supplementary Fig. 8). With the few sorted pooled splenic B cells, we confirmed the downregulation of *Bcl2* mRNA and BCL2 protein in pre-malignant B cells after LFPRLR knockdown (Fig. 4f, g).

In contrast to SLE-prone mice, bone marrow B-cell counts in DLBCL-prone mice were significantly increased after LFPRLR knockdown. However, their expression of BCL2 was again significantly reduced (Fig. 4h, i). Together with reduced splenic B-cell numbers (Fig. 4d), this suggests that LFPRLR knockdown impairs the dissemination of the pre-malignant bone marrow B cells into the periphery.

Overall, the PRL-LFPRLR axis induces B-cell lymphocytosis and promotes the expression of TCL1 and BCL2 oncoproteins in pre-malignant B cells, thereby raising the chance of overt B-lymphomagenesis (Fig. 4j).

**PRL-LFPRLR signaling maintains overt human B-cell malignancies.** Circulating and local PRL has been shown to be elevated in malignant lymphomas[55,56]. However, the consequences of the local and systemic overproduction of PRL in lymphomas are unknown. We determined whether expression of autocrine/paracrine PRL in tumor samples from patients with DLBCL and/or Burkitt's lymphoma (BL)[57–59] was indicative of clinical outcome. Higher-than-median expression of *PRL* transcript (*PRL*^high) significantly associated with lower overall survival in patients with DLBCL/BL, both at diagnosis[57] and after standard treatments[58,59] (Fig. 5a). Because expression of individual PRLR isoforms in these samples was unavailable, we compared overall survival after separating patients into *PRLR*^high and *PRLR*^low groups based on their median mRNA expression of total *PRLR*. Total *PRLR* levels did not correlate with survival of DLBCL/BL patients (Supplementary Fig. 9), suggesting either that local production of autocrine PRL was more important than levels of PRLR expression and/or that some tumors still express some level of SFPRLRs.

Next, we investigated whether B-lymphoblasts in ALL patients with poorly prognostic MYC/BCL2-driven B-ALL[60] express more autocrine PRL than their healthy counterparts. Unlike normal B cells, B cells in B-ALL patients produced autocrine *PRL* (Fig. 5b). While normal B cells expressed both *LF/IF* and *SFs*, 50% of the patients with MYC/BCL2-driven B-ALL expressed only the *LF/IFPRLR* (Supplementary Fig. 10). Consistent with these findings, unlike MYC/BCL2-driven B-ALL (VAL) and DLBCL

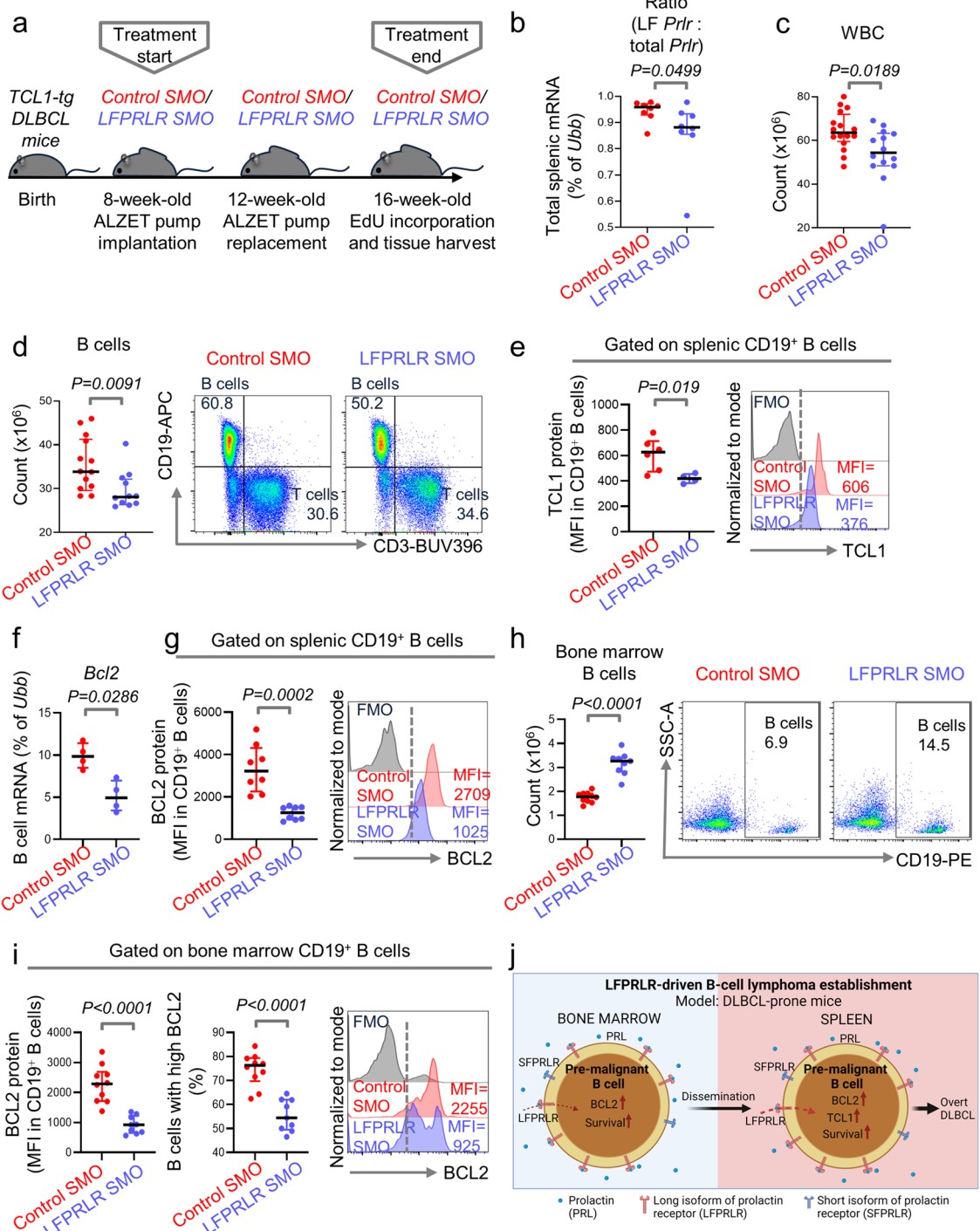

**Fig. 4 Knockdown of LFPRLR in DLBCL-prone mice suppresses factors that can drive overt B-cell lymphomagenesis. a–c** Experimental design (**a**), verification of LFPRLR knockdown in total splenic WBCs by qPCR (**b**), and splenic WBC counts (**c**) in *TCL1-tg* DLBCL-prone mice treated with control SMO [$n = 8$ for (**b**) and $n = 16$ for (**c**), red] or LFPRLR SMO [$n = 8$ for (**b**) and $n = 14$ for (**c**), blue]. **d–i** Quantitation and representative flow cytometry plots of CD3⁻CD19⁺ B cells (**d, h**), splenic B cell-specific expression of TCL1 (**e**), and B cell-specific expression of BCL2 (**f, g, i**) in spleen and bone marrow of *TCL1-tg* mice treated with control SMO [$n = 14$ for (**d**), $n = 6$ for (**e**), $n = 4$ for (**f**), $n = 8$ for (**g**) and $n = 10$ for (**h, i**), red] or LFPRLR SMO [$n = 12$ for (**d**), $n = 4$ for (**e**), $n = 4$ for (**f**), $n = 8$ for (**g**) and $n = 9$ for (**h, i**), blue]. One representative histogram from each group for measurement of TCL1 protein in spleen and BCL2 protein in spleen and bone marrow is shown. **j** PRL-LFPRLR signaling upregulates BCL2 and increases survival of pre-malignant B cells, thereby enhancing the risk of establishment of overt B-cell malignancies. Graphs show median ± interquartile range. Each dot in qPCR analyses of transcripts corresponds to the mean expression of that gene in pooled samples from 3 mice that were run in 3 technical replicates. *Ubb* was used as the house keeping gene in qPCR. Exact *p*-values were calculated using the Mann–Whitney *U* test. ns non-significant, FMO fluorescence minus one control, MFI median fluorescence intensity. **j** was created with BioRender.com.

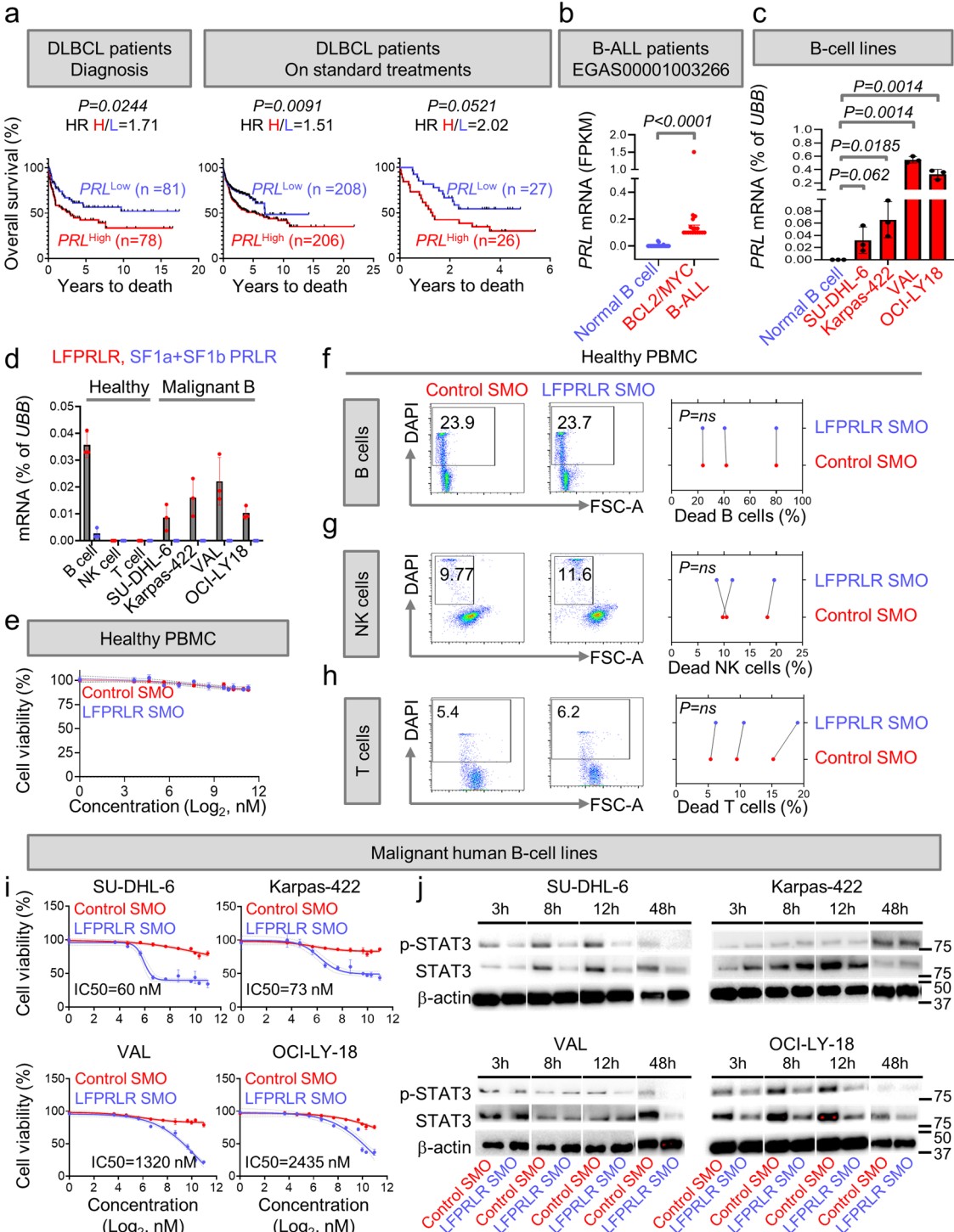

(Karpas-422, SU-DHL-6, OCI-LY18) human cell lines, normal B cells of healthy donors did not express autocrine *PRL* and expressed the counteractive *SFPRLRs* (Fig. 5c, d). We infer that the upregulation of PRL-LF/IFPRLR signaling (high *PRL* expression and no *SFPRLR*) likely contributes to the maintenance of aggressive B-cell malignancies.

The preclinical potential of the LFPRLR SMO was confirmed by showing that LFPRLR knockdown is not toxic to healthy human peripheral blood mononuclear cells (PBMCs) (Fig. 5e). Although normal B cells express more *LF* than *SFPRLRs* (Fig. 5d), SFPRLRs do not have to be present in a 1:1 ratio to inhibit the activity of LFPRLRs[23,25,26]. Moreover, the lack of production of

autocrine *PRL* by normal B cells (Fig. 5b, c) likely renders them insensitive to LFPRLR knockdown (Fig. 5f). Unlike B cells, normal T and NK cells did not express the *LFPRLR* (Fig. 5d) and, therefore, administration of the LFPRLR SMO was not toxic to these cells (Fig. 5g, h, Supplementary Fig. 11).

We measured malignant B-cell survival after LFPRLR SMO treatment by 3-(4,5-dimethylthiazol-2-yl)−5-(3-carboxymethox-yphenyl)−2-(4-sulfophenyl)−2H-tetrazolium (MTS) assay and flow cytometry. Because malignant B cells produce PRL that can drive their PRLR signaling, we did not add exogenous PRL in our experiments. LFPRLR knockdown killed Karpas-422 and SU-DHL-6 DLBCLs with half maximal inhibitory concentration

**Fig. 5 PRL-LFPRLR signaling maintains overt human B-cell malignancies. a** Overall survival probabilities of patients with B-cell lymphomas from 3 datasets (GSE4475, GSE10846, and E-TABM-346) divided into *PRL*^high and *PRL*^low based on the median expression of *PRL* mRNA. GSE4475 includes 123 DLBCL and 36 Burkitt lymphoma patients at diagnosis. GSE10846 includes samples from 414 DLBCL patients collected after treatment with CHOP (cyclophosphamide, hydroxydaunorubicin, vincristine, and prednisone) or rituximab-CHOP, and E-TABM-346 includes samples from 53 DLBCL patients collected after treatment with rituximab-CHOP or CHOP. HR Hazard Ratio. **b, c** Expression of *PRL* mRNA in CD3⁻CD19⁺ B cells from healthy donors ($n = 27$) and B-lymphoblasts of patients with high-grade MYC/BCL2-driven B-ALL ($n = 18$) by whole RNA sequencing (EGAS00001003266, St. Jude, median ± interquartile range, *p*-value calculated by Mann–Whitney *U* test) (**b**); and in CD3⁻CD19⁺ B cells isolated from PBMC of healthy donors ($n = 3$) and in malignant human B-cell lines (Karpas-422, SU-DHL-6, VAL and OCI-LY18, (**c**)) by qPCR. **d** Expression of *LFPRLR* (red dots) and *SFPRLR* (blue dots) mRNA in CD3⁻CD19⁺ B cells, CD3⁻CD56⁺ NK cells and CD19⁻CD3⁺ T cells sorted from PBMC of healthy donors ($n = 3$) and malignant B-cell lines (Karpas-422, SU-DHL-6, VAL and OCI-LY18) by qPCR. **e–h** MTS assay to compare viability of PBMC isolated from healthy donors after treatment with either control SMO or LFPRLR SMO for 48 h (**e**), and flow cytometry to compare the viability (% FSC^highDAPI⁻) of MACS-sorted CD3⁻CD19⁺ B cells (**f**), CD3⁻CD56⁺ NK cells (**g**), and CD3⁺ T cells (**h**) isolated from PBMC of four healthy donors after treatment with 73 nM of either control SMO or LFPRLR SMO for 48 h. **i** Viable cell number was assessed using MTS assay in malignant human B-cell lines after treatment with control SMO or LFPRLR SMO for 48 h. **j** Immunoblotting showing reduction in active phosphorylated STAT3 and global STAT3 levels in three out of four malignant B cell lines treated at their IC50 concentration of LFPRLR SMO for 3, 8, 12 and 48 h. β-actin was used as the loading control in immunoblotting. *P* values for survival curves were measured by log-rank (Mantel–Cox) test. For all other experiments, *p*-values were calculated by unpaired *t* test from 3 independent experiments/biological replicates (±SEM), unless otherwise indicated. qPCR of each biological sample was conducted in three technical replicates. *UBB* was used as the house-keeping gene in qPCR. For all flow cytometry-based measurements of cell death, the IC50 concentration of the Karpas-422 line was used (73 nM). FPKM = fragments per kilobase of transcript per million mapped reads.

(IC50) at 60–70 nM of LFPRLR SMO; however, IC50s for the VAL B-ALL and OCI-LY18 DLBCL cell lines were 20–40-fold higher (Fig. 5i, Supplementary Fig. 12). Hence, human B-cell malignancies are differentially sensitive to knockdown of the LF/IFPRLR.

Next, we investigated the mechanisms by which LFPRLR knockdown reduces the fitness of malignant B cells. To determine whether suppression of STAT signaling is a potential mechanism of action of the LFPRLR SMO, we compared expression of the 'STAT-activating' LFPRLR and the 'non-STAT-activating' IF PRLR[23] relative to one another and their absolute expressions before and after LFPRLR knockdown. As expected, expression of both *LF* and *IF PRLR* decreased after treatment with LFPRLR SMO. However, *LFPRLR* expression was ~10–80- fold higher than *IF* expression in each cell line in the absence of treatment, suggesting that LFPRLR SMO may kill malignant lymphocytes by reducing the opportunity for STAT activation by PRL (Supplementary Fig. 13). As predicted, phosphorylation (activation) and production of STAT3 decreased after LFPRLR knockdown in three out of four malignant B cell lines treated with their cell death inducing IC50 concentrations of LFPRLR SMO (Fig.5j, Supplementary Fig. 14). pSTAT5 was undetectable in all cell lines, although levels of global STAT5 were reduced in some lines after treatment with LFPRLR SMO (Supplementary Fig. 15). Our results are consistent with the recent finding that STAT3, and not STAT5, is activated downstream of PRLR in autoimmune B cells[18]. We conclude that activation of STAT3 downstream from LFPRLR may play an important role in sustaining of B-cell malignancies.

**Knockdown of LFPRLR reduces progression of overt human DLBCL in vivo.** To further demonstrate that knockdown of LFPRLR could be therapeutically beneficial by inhibiting progression of overt B-cell malignancies, we tested the efficacy of LFPRLR SMO in the clearance of two human DLBCL cell-line derived xenografts (CDX) transplanted into immune-deficient *NOD-SCID IL2Rγ⁻/⁻* (NSG) mice. Control SMO or LFPRLR SMO were continuously delivered via alzet minipump for 15 days (Fig. 6a). We confirmed that LFPRLR was significantly knocked down in tumors isolated from mice treated with LFPRLR SMO (Fig. 6b). Consistent with our short-term in vitro studies (Fig. 5i), we observed that LFPRLR knockdown significantly slowed DLBCL progression in vivo in the two independent CDX models used (Fig. 6c, d). We conclude that the knockdown of LFPRLR

effectively reduces the progression of overt B-cell malignancies. Because the human LFPRLR SMO does not affect mouse LFPRLR[61] we also conclude that this result stems from a direct effect on the B-cell lymphomas. At the same dose of transplanted lymphoma cells, the tumors derived from SU-DHL-6 were more sensitive to LFPRLR knockdown than those derived from OCI-LY18, consistent with the in vitro results (Figs. 5i and 6c, d).

**Knockdown of LFPRLR in malignant human B cells reduces BCL2 and MYC expression.** Given the differential sensitivity of the cell lines to LF/IFPRLR knockdown both in vitro and in vivo, we hypothesized that the intrinsic properties of malignant B cells, including their level of autocrine PRL secretion and expression of oncogenic drivers could dictate their sensitivity to increasing concentrations of LFPRLR SMO.

First, we determined whether differential secretion of PRL from malignant B cells was related to their sensitivity to LFPRLR knockdown. Cells sensitive to nM concentrations of LFPRLR SMO (SU-DHL-6 and Karpas-422, Fig. 5i) secreted much higher levels of PRL than their counterparts sensitive only to μM concentrations (VAL and OCI-LY-18, Fig. 5i) (Fig.7a). Importantly, neutralization of the secreted PRL with rabbit anti-PRL, even at low concentrations, markedly reduced the viability of cell lines with high amounts of PRL secretion and nM sensitivity to LFPRLR SMO (Fig. 7b, Supplementary Fig. 16). In contrast, in malignant B cells sensitive to μM concentrations of LFPRLR SMO, significant impairment of cellular fitness was only observed at high concentrations of anti-PRL (Fig. 7b, Supplementary Fig. 16). Hence, malignant B cells' dependence on LFPRLR positively correlates with their secretion of autocrine PRL.

We investigated other possible mechanisms that could explain differential sensitivity of malignant B cells to LFPRLR SMO treatment. Of note, treatment with low nanomolar concentrations of LFPRLR SMO significantly reduce *LFPRLR* expression in SU-DHL-6 and Karpas-422, but not in VAL and OCI-LY18 cell lines. (Fig. 7c, d). These findings suggest dysregulation of the splicing factors required for optimal effect of the LFPRLR SMO in the less sensitive VAL and OCI-LY18 lines.

Splicing dysregulation is often caused by constitutive over-expression of MYC[62], an important downstream target of PRL signaling that co-operates with BCL2[63], another downstream target of PRL[16–19], to sustain B-cell malignancies. A striking difference between the LFPRLR SMO-sensitive SU-DHL-6 and Karpas-422 and the less sensitive VAL and OCI-LY18 lines is the

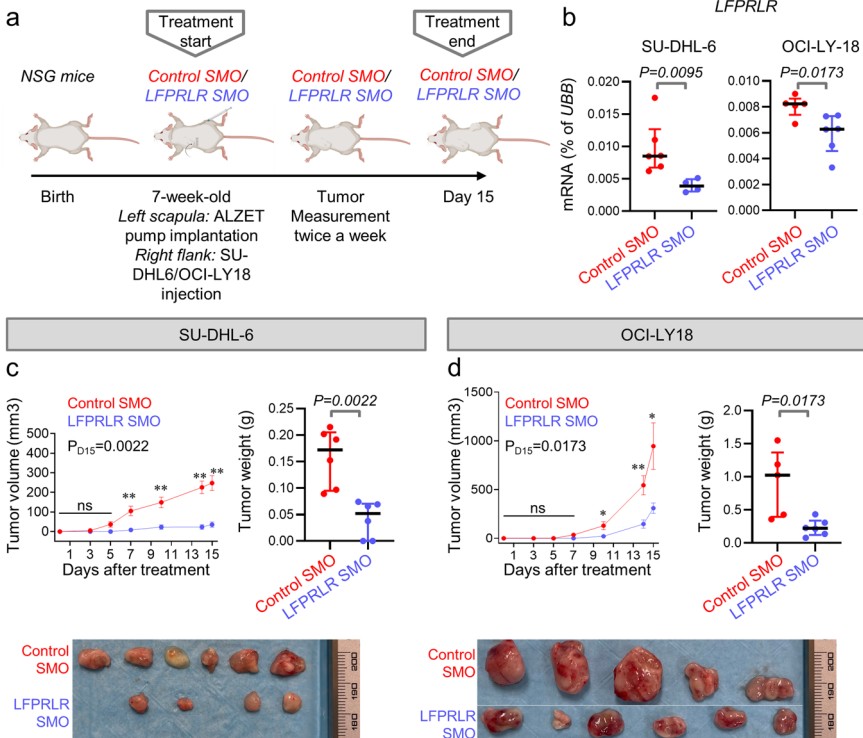

**Fig. 6 Knockdown of LFPRLR reduces progression of overt human DLBCL in vivo. a** Experimental design for measuring the in vivo efficacy of LFPRLR SMO in reducing progression of overt human DLBCL in malignant B-cell line derived xenograft (CDX) models. **b** qPCR validation of LFPRLR knock down in tumors isolated from euthanized transplant recipient mice treated with control SMO ($n = 6$ for SU-DHL-6 and $n = 5$ for OCI-LY-18, red) or LFPRLR SMO ($n = 4$ for SU-DHL-6 and $n = 6$ for OCI-LY-18, blue) on day 15. Each dot in qPCR analyses of transcripts corresponds to the mean expression of that gene in tumors from one CDX recipient mouse calculated from 3 technical replicates. *UBB* was used as the house-keeping gene in qPCR. **c, d** Change in tumor volume and tumor weight on the day of euthanasia (day 15) in *NSG* mice that received subcutaneous injections of $7 \times 10^6$ malignant human B cells [SU-DHL-6 (**c**) and OCI-LY18 (**d**)] and were treated with control SMO ($n = 6$ for SU-DHL-6 and $n = 5$ for OCI-LY18, red) or LFPRLR SMO ($n = 6$ for SU-DHL-6 and $n = 6$ for OCI-LY18, blue). Tumor volume panels in (**c, d**) show mean ± SEM. All other graphs show median ± interquartile range. Exact p-values were calculated using the Mann–Whitney U test. ns non-significant, *$0.01 < p < 0.05$, **$0.001 < p < 0.01$. **a** was created with BioRender.com.

constitutive expression of MYC in addition to BCL2 due to chromosomal translocations in VAL and OCI-LY18. Therefore, we determined whether an inability to downregulate MYC and/or BCL2 in VAL and OCI-LY18 reduced their sensitivity to low concentrations of LFPRLR SMO. As predicted, treatment with a low concentration (73 nM) of LFPRLR SMO reduced MYC protein and *BCL2* mRNA and BCL2 protein in the sensitive SU-DHL-6 and Karpas-422 lines, but such reduction was absent in the VAL and OCI-LY18 lines (Fig. 7c–f, Supplementary Fig. 17). We conclude that B-cell malignancies rely on PRL signaling to different extents, with some indication that dual translocations of *MYC* and *BCL2* override this reliance (Fig. 7g).

## Discussion

Our newly identified causal role of the PRL-LF/IFPRLR axis in promoting the evolution of B-cell malignancies suggests that the LF/IFPRLR may represent a therapeutic target in patients with B-cell malignancies and in those vulnerable to developing these malignancies.

The molecular mechanisms underlying the increased risk of transformation of B-cell clones in autoimmune diseases are not well understood[5,64]. We find that LFPRLR not only promotes the accumulation of deleterious B cells by increasing their proliferation and their chances of being mutated by AID, but also exacerbates B-cell pathology in SLE. Thus, targeting the synthesis of the LF/IFPRLR represents an attractive strategy to lower the risk of SLE-associated B-lymphomagenesis.

In pre-malignant B-cells, the LFPRLR promotes expression of BCL2 and TCL1, two oncoproteins whose high levels have been found to correlate with poor clinical prognosis in patients with B-cell lymphomas[65–67]. Hence, our studies underscore the importance of LFPRLR in driving overt B-cell lymphomagenesis. Given the limited attempts to interfere with cancer progression from a pre-malignant state to overt cancer, our studies support the importance of exploring the potential of the LFPRLR SMO in preventing overt leukemogenesis/ lymphomagenesis from pre-leukemic[8] and indolent B cells[7] and monoclonal B-cell lymphocytosis[68].

We find that PRL-LF/IFPRLR signaling maintains the growth and survival of established overt human B-cell malignancies in vitro and in vivo, and that this maintenance is associated with the induction of p-STAT3, MYC, and BCL2. The LFPRLR SMO may thus be a non-toxic alternative to prevent the activation of STAT3, and/or block the production of BCL2 and the 'difficult-to-drug' MYC oncoprotein[69]. Our findings showing that normal T and NK cells are insensitive to LF/IFPRLR knockdown also suggest that LFPRLR SMO may be combined with immune cell-based therapies for treating B-cell malignancies.

The LFPRLR SMO offers several advantages over other drugs that affect PRL signaling. For example, dopamine agonists that inhibit PRL production by the pituitary[33,70] have no effect on non-pituitary autocrine/paracrine sources of PRL in the body[12], including malignant B cells. Dopamine agonists would also abrogate beneficial PRL-SFPRLR signaling, which may mediate apoptosis of deleterious B cells. By isoform-specific targeting, the

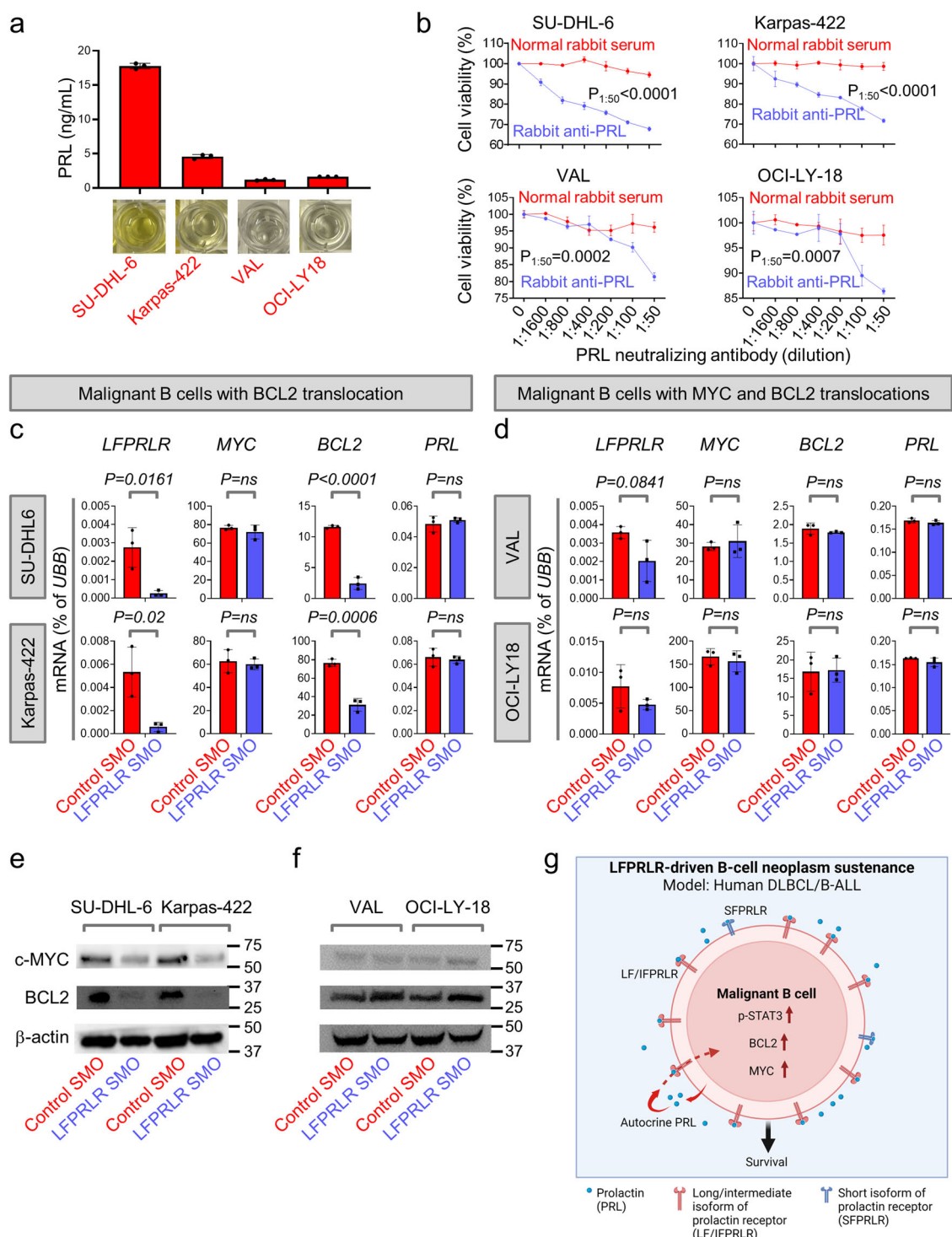

chances of off-target toxicities are reduced, such as in the liver, which primarily expresses the SFPRLRs[71]. Traditional receptor antagonists or anti-PRLR antibodies affect all PRLR isoforms since they all have identical extracellular domains[18]. Therefore, unlike the LFPRLR SMO, they cannot specifically target the axis governing disease evolution.

Our goal was to determine whether the LFPRLR is causal in B-cell malignancy evolution by examining mouse models exhibiting increased risk of initiation or enhanced progression of this disease, and human data. Each murine model has its limitations. For example, death from SLE in control SMO-treated *Mrl-lpr* mice[27] precludes survival studies that determine the

extent to which LFPRLR knockdown blocks the initiation of B-cell lymphomas. In *TCL1-tg* mice, development of overt DLBCL is slow and uneven[28], demanding survival studies with a large number of mice and of more than a year's duration. Therefore, in both models, we instead examined how LFPRLR promotes the expression of well-known molecular drivers of B-lymphomagenesis specific to the age of the mice at the time of treatment completion with the SMOs. Our findings demonstrating a role for LFPRLR signaling in three types of dysregulated B cells (non-malignant but aberrant, pre-malignant, and malignant) strongly support further preclinical development of the LFPRLR SMO.

**Fig. 7 Knockdown of LFPRLR in malignant human B cells reduces BCL2 and MYC expression. a** PRL secretion level by ELISA and **b** viable cell number after treatment with rabbit anti-PRL or rabbit normal serum, assessed by MTS assay in malignant human B-cell lines. *P*-values by unpaired *t* test at different concentrations of anti-PRL or normal serum in **b**: SU-DHL-6 ($P = 0.0009$ in 1:1600 and $P < 0.0001$ in 1:800, 1:400, 1:200, 1:100 and 1:50), Karpas-422 ($P = 0.0283$ in 1:1600, $P = 0.0016$ in 1:800, $P = 0.0004$ in 1:200 and $P < 0.0001$ in 1:400, 1:100 and 1:50), VAL ($P = 0.0066$ in 1:1600, $P = 0.0178$ in 1:100 and $P = 0.0002$ in 1:50) and OCI-LY18 ($P = 0.0253$ in 1:1600, $P = 0.0007$ in 1:800, $P = 0.0051$ in 1:100, $P = 0.0007$ in 1:50). *P*-values were calculated by unpaired *t* test (±SD). *P* values for 1:50 are indicated in the figure. ns non-significant. **c, d** *LFPRLR* mRNA levels and expression of *MYC*, *BCL2* and *PRL* transcripts in malignant human B-cell lines (Karpas-422, SU-DHL-6, VAL and OCI-LY18) treated with 73 nM of control SMO or LFPRLR SMO for 48 h. Each qPCR sample was run in 3 technical replicates. One representative of 3 independent experiments is shown. *UBB* was used as the house keeping gene in qPCR. *P*-values are calculated by unpaired *t* test from 3 technical replicates (±SEM). Cell lines highly sensitive to low concentration (73 nM) of LFPRLR SMO are shown in (**c**) and cell lines less sensitive to LFPRLR SMO at 73 nM are shown in **d**. **e, f** Immunoblotting to compare the expression of MYC and BCL2 proteins in human DLBCL cell lines (Karpas-422, SU-DHL-6, VAL, and OCI-LY18) treated with 73 nM of control SMO or LFPRLR SMO for 48 h. One representative of 3 independent experiments is shown. β-actin was used as the loading control in immunoblotting. Cell lines highly sensitive to low concentration (73 nM) of LFPRLR SMO are shown in **e** and cell lines less sensitive to LFPRLR SMO at 73 nM are shown in **f**. **g** Signaling of PRL through the LF/IFPRLR induces the expression of MYC and BCL2 in overt human DLBCL and B-ALL, thereby promoting malignant B-cell growth and survival. LFPRLR SMO thus represents a promising candidate to treat these malignancies. **g** was created with BioRender.com.

## Methods

**SMO**. SMO and control oligomers linked to octaguanidine dendrimers for cell/tissue penetration in vivo, were custom synthesized (Gene Tools, Philomath, Oregon) (Supplementary Table 1a). SMO sequences were designed to bind to the intron 9-exon 10 junction in mouse or human *LF/IFPRLR* pre-mRNA[29].

**Animal models**. Animal studies were conducted in compliance with Institutional Animal Care and Use Committees (IACUC) at City of Hope and University of California, Riverside. 6-week-old mice female SLE-prone *MRL-lpr* mice homozygous for the Fas cell surface death receptor mutation (*Fas^lpr*) were anesthetized with isoflurane, and Alzet minipumps (Durect, Cupertino, CA) were implanted subcutaneously between the scapulae. Mice were randomly assigned to control SMO or LFPRLR SMO groups and coded by ear punch. Animals in each group were housed individually after pump implantation until wound clips were removed. Alzet pumps that delivered 100 pmoles/h/mouse of either control SMO or LFPRLR SMO were changed after 4 weeks. At week 8 of treatment, two hours before euthanasia, each animal received an intraperitoneal injection of 2.8 mg of the nucleoside analog, 5-ethynyl-2′-deoxyuridine (EdU). 8-week-old male and female DLBCL-prone *TCL1-tg* mice were implanted with Alzet minipumps and treated with control or LFPRLR SMO for 8 weeks, as described for *MRL-lpr* mice. Immune-deficient *NSG* mice were used as transplant recipients in the human B-cell malignancy CDX models. Treatment with SMOs in the *NSG* CDX-recipient mice was conducted as described above for the primary mouse models of SLE and DLBCL, except that the treatment duration was shorter to ensure tumor size remained within guidelines in the control SMO treated animals. Transplant recipient CDX mice were euthanized following the stringent guidelines on tumor volume and health of mice laid down by IACUC of City of Hope.

**Tissue processing**. Spleens were placed in Roswell Park Memorial Institute (RPMI) medium containing 40 units/ml DNAse (RPMI+), transferred to a 70 μM cell strainer over a new dish, and mashed with a syringe plunger. Bone marrow was flushed from femurs and tibia with RPMI+ using a syringe with a 25½ G needle into a 70 μM cell strainer. A syringe plunger broke up cell clumps and residual cells were washed through the strainer. After pelleting and red blood cell lysis in 1X lysis buffer (BD Biosciences), WBCs were washed in RPMI+. To ensure that all samples across different groups were treated identically, cells were resuspended in freezing solution (90% FBS/10% DMSO) and stored in liquid nitrogen until analysis.

**Cell lines and culture**. Mycoplasma-negative human cell lines were obtained from Deutsche Sammlung von Mikroorganismen und Zellkulturen and cultured in RPMI 1640 with 10% fetal bovine serum, 100 U/mL Penicillin, and 100 μg/mL Streptomycin (Complete RPMI) (Invitrogen/Life technologies). Because human PRLRs do not respond to non-human PRL[61], bovine PRL in the FBS-supplemented media would not influence the human lymphoma cell lines used.

**Quantitative real-time PCR (qPCR)**. RNA was extracted using a kit (NucleoSpin® RNA Plus, MACHERY-NAGEL) according to manufacturer's instructions. cDNA was synthesized using SuperScript™ IV Reverse Transcriptase (Invitrogen). qPCR was conducted using Power SYBR Green PCR master mix (ThermoFisher Scientific) and QuantStudio 7 flex real-time PCR system (Applied Biosystems). Supplementary Table 1b lists qPCR primers.

Total murine *Prlr* = LF + SF3 (SF1 and SF2 were not expressed in splenocytes and B cells). Total human *PRLR* = LF + IF + SF1a + SF1b. There is a huge degree of variability in the absolute expression of the isoforms across individual mice in each strain. To account for this variability and to most accurately predict changes in pro-proliferative, anti-apoptotic effects of PRLRs in cells after LFPRLR knockdown, expression of *PRLR* isoforms in each mouse was represented as the ratio of *LFPRLR*: total *PRLR*. An exception to calculating the ratio is made in instances where the cells express only *LF/IFPRLR*, as in some malignant human B cells.

**B-cell clonality**. *IGH* repertoire of splenic B cells were PCR-amplified using primers for the J558 V$_H$-region gene and the Cμ constant region (Supplementary Table 1c), as described previously[72,73]. Selective amplification of the J558 variable region of the *IGH* (V$_H$) family was conducted due to the increased representation of this family in both the production of autoantibodies and non-binding antibodies in *MRL-lpr* SLE-prone mice[74]. Amplicons were subjected to next generation sequencing (NGS): TrueSeq library was prepared by adapter ligation to full-length products and sequenced on the MiSeq Illumina 2 × 150 bp platform. Abundances of unique nucleotide sequences were calculated and CDR3 length, spectrum, and functionality were analyzed using International Immunogenetics Information System (IMGT) HighV-QUEST. VDJTools (v1.1.7) was used for visualization. CDR3 amino acid frequencies were calculated using customized R scripts and heatmaps were generated using the pheatmap package and unsupervised hierarchical clustering using Euclidian distance. NGS data are available in GSE207186.

**Analysis of PRLR isoforms in B-ALL patients**. Expression of human *PRLR* isoforms was parsed based on previous knowledge that *SFs* lack most of exon 10, whereas *LF* and *IF* include entire or large portions of exon 10[21]. Raw paired-end fastq files of RNA-seq were mapped to human GRCh38 using STAR v.2.7.6a. Raw exon counts across the whole genome were obtained using DEXSeq[75], normalized based on variance stabilizing transformation method using R package DESeq2[76], and batch corrected using R package sva.

**Flow cytometry**. Cells were thawed in Complete RPMI and stained with fluorochrome-tagged surface antibodies and Ghost-UV450 for 30 min on ice. Using eBioscience™ Transcription Factor Staining Buffer Set, cells were fixed, permeabilized, and stained with intracellular antibodies for 30 min on ice followed by acquisition on the BD FACSymphony cytometer. Gates were set using single-stained and fluorescence minus one (FMO) controls. Data were analyzed using FlowJo10.7.1. Supplementary Table 2a lists cytometry antibodies.

**Cell cycle**. EdU incorporation was assessed using Click-iT™ Plus EdU flow cytometry. Cells were washed in 1% bovine serum albumin in phosphate buffered saline and stained for surface antigens, as above, followed by fixation, permeabilization, click-iT EdU detection, and 4′,6-diamidino-2-phenylindole (DAPI) staining. Cells were analyzed on BD FACSymphony and FlowJo10.7.1.

**Magnetic activated cell sorting (MACS)**. Mouse splenic B cells were enriched using CD19-microbeads (Miltenyi). Human healthy donor peripheral blood mononuclear B, T, and NK cells (PBMCs) were enriched using Miltenyi kits. Enrichment was confirmed by flow cytometry.

**Immunoblotting**. Cells were lysed in RIPA buffer (Sigma) supplemented with 1% protease inhibitor 'cocktail' (Pierce). Proteins were separated by electrophoresis through 4–20% TGX gradient gels (BioRad) and transferred to polyvinylidene fluoride membranes (Immobilon; Millipore). Mouse and human proteins were detected by immunoblotting using antibodies (Supplementary Table 2b) and western enhanced chemiluminescence (BioRad).

**MTS Assay**. One thousand cells were seeded per well in 96-well plates and incubated in Complete RPMI containing different concentrations of control or LFPRLR SMO (0, 5, 25, 125, 250, 500, 750, 1000, 1500, 2000 nM) at 37 °C for 48 h.

Twenty μl of MTS (Promega) was added to 100 μl of media containing cells and incubated for 3 h. Absorbance was measured at 490 nm with TECAN Infinite M1000 Pro microplate reader.

**Enzyme linked immunosorbent assays (ELISA).** For the quantitative determination of human PRL concentrations in cell supernatant, we used Biotechne Quantikine ELISA kit Cat #DPRL00. According to manufacturer's instructions, we measured secreted PRL in cells supernatant based on standard curve plotted using different concentrations of human recombinant PRL. Sensitivity of human PRL ELISA kit is 0.264 ng/mL.

Anti-dsDNA was assayed by ELISA using a kit from Chondrex (#3031). To ensure consistent values between assays, we diluted and then stored samples in assay dilution buffer for >24 h at −20 °C before assay.

**PRL neutralization assay.** Ten thousand cells were seeded per well in 96-well plates and incubated in Complete RPMI containing different concentrations of normal rabbit serum or rabbit anti-PRL (NIDDK standard, AFP55762089) (0, 1:1600, 1:800, 1:400, 1:200, 1:100, 1:50 dilution) at 37 °C for 48 h. Twenty μl of MTS (Promega) was added to 100 μl of media containing cells and incubated for 3 h. Absorbance was measured at 490 nm with a TECAN Infinite M1000 Pro microplate reader.

**LFPRLR SMO treatment efficacy studies in CDX models of human B-cell malignancies.** Six female immune-deficient *NSG* mice (Jackson Laboratories) received subcutaneous injections of $7 \times 10^6$ malignant B cells suspended in 200 μL matrigel (BD Biosciences) in their right flank. Cell lines were pre-treated in vitro for 3 h with either control SMO or LFPRLR SMO at their respective IC50 concentrations. Alzet minipumps delivering the SMOs were implanted on the left scapula of each mouse. Tumor burden was assessed twice weekly by caliper measurement. Tumor volume was calculated using the formula: V = ½ (length × width$^2$). Mice were euthanized after 15 days of treatment.

**Statistics and reproducibility.** Exact p-values are provided if significant ($P < 0.05$) or trending towards significance ($0.05 < P < 0.1$). Statistical tests were two-tailed. Pairwise comparisons between mouse cohorts were performed using the Mann–Whitney U test. For Kaplan–Meier survival analyses, p-values were calculated using a log-rank test. Female SLE-prone mice were used because SLE is more frequent in females. In experiments involving pre-malignant or malignant B cells, we represented both sexes equally because B-cell malignancies develop at the same frequencies in both males and females and malignant B cells from males and females had similar PRL and PRLR isoform expression. qPCR samples were run in 3 technical replicates. Experiments involving cell lines were repeated 3 times to determine reproducibility.

**Reporting summary.** Further information on research design is available in the Nature Portfolio Reporting Summary linked to this article.

## Data availability

Next generation sequencing of the immunoglobulin heavy chain repertoire described in this study is available in Gene Expression Omnibus under accession GSE207186. All source data for main figures in the manuscript are provided as supplementary Data 1. Uncropped and unedited immunoblots are provided in Supplementary Figures.

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

## Acknowledgements

We thank Dr. Alexey Danilov (City of Hope) for providing us with human DLBCL and B-ALL cell lines. We thank Dr. Nora Heisterkamp and Dr. Rui Su (City of Hope) for their productive discussions. Schematics were created using BioRender.com.

This work was supported by the following grants: City of Hope-University of California Riverside Biomedical Research Initiative Award (S.S., A.M.W.), Research Start-Up Budget from the Beckman Research Institute of the City of Hope (S.S.), Junior Investigator Research Development Award from the Circle of Service Foundation (S.S.), P50 CA107399-12 City of Hope Lymphoma SPORE Career Enhancement Program Pilot Award (S.S.) from the National Cancer Institute of the National Institutes of Health [PIs: Stephen J. Forman (S.J.F.), Larry Kwak], a Shared Resources Pilot Grant from City of Hope (S.S.), a Conquer Cancer Now Award from the Concern Foundation (S.S.), and a Congressional Families Program Award from the Prevent Cancer Foundation (S.S.). S.S. is a Scholar of the American Society of Hematology and is supported in part by a Translational Research Program Grant (LLS 6624-21) from the Leukemia and Lymphoma Society. Research reported in this publication included work performed in the Analytical Cytometry Core and Integrative Genomics Core Shared Resource supported by the National Cancer Institute of the National Institutes of Health under grant number P30CA033572. The content is solely the responsibility of the authors and does not necessarily represent the official views of the National Institutes of Health.

## Author contributions

A.T.K. conducted most of the experiments, analyzed data, and interpreted the results. A.K. assisted A.T.K. with flow cytometry experiments, data analyses, and interpretation. A.S.O., K.C.R., S.A., S.J.L., A.M.W., and M.Y.L. conducted experiments. Z.H., B.D., and Z.G. developed scripts to parse the expression of specific PRLR isoforms in RNA sequencing data from patients with high-risk B-ALL. X.W. developed scripts to analyze next generation sequencing data of the B-cell repertoire in SLE-prone mice. W.S., I.S., and E.M. provided guidance on B-cell pathology in SLE. M.M., S.J.F., and J.L.K. provided scientific advice in B-cell malignancy pathogenesis and transition of autoimmune diseases to B-cell malignancies. A.M.W. provided expert advice on PRL signaling. S.S. and A.M.W. conceived the study, developed the experimental methodology, interpreted the results, provided administrative, technical, and material support and supervised the study. S.S. wrote the manuscript. All authors reviewed and edited the manuscript.

## Competing interests

A.M.W. and S.S. declare the following competing interests: The work described in this study is covered under a pending US patent application (Inventors: A.M.W. and S.S.). All other authors declare no competing interests.
