## [Peer Review File · Communications Biology]

Reviewers' comments:

Reviewer #1 (Remarks to the Author):

Although early studies of prolactin on the immune system did not reveal strong activity in health, its actions are increasingly recognized in the context of underlying pathology. Here the authors investigate prolactin signals in B cell malignancies, employing mouse models, and human data sets, samples and cell lines. They utilize an innovative approach to reduce canonical prolactin signals by modulating splicing of prolactin receptor transcripts, and investigate the consequences in in vivo mouse models of systemic lupus erythematosus (SLE) and B cell lymphoma. A relatively small change in ratio of LFPRLR/ total PRLR mRNAs in splenic leucocytes reduces the number of B but not T cells and plasmacytoid dendritic cells in the SLE model, associated with reduced cell cycle progression and reduced Bcl2 and Aid mRNA in splenic B cells, both recognized players in clinical disease. In the DLBCL model, this treatment reduces B cells in the spleen, although it increases them in the bone marrow. They explore a role in human clinical disease by interrogating publicly available data sets of patient DLBCL and ALL tumors, identifying an association of high expression of prolactin with a worse prognosis. Notably, prolactin mRNA is very low in B cells of healthy patients, but is much higher in patient tumors and several B cell tumor cell lines. In vitro studies in multiple human B cell tumor lines indicate that prolactin plays a role in cell lines with BCL2 translocations, but not both MYC/BCL2 translocations.

These studies in multiple models underscore the importance of canonical prolactin signals in various states of B cell pathology, and emphasize the changing expression of components of this pathway in disease. Their studies identify specific sites of action of prolactin, both physical location and relevant molecular targets. Together, their findings illuminate a role for prolactin in both increasing the risk for B cell pathologies and in driving a subset of DLBCL tumors, and suggest characteristics of patient tumors that may respond to directed therapies. Addressing the following would be helpful:

1. It is interesting that very substantial effects on biology are initiated by a relatively small, although significant, reduction in the ratio of long form PRLR mRNA, compared to total PRLR mRNA. It is shown in human cells (Fig. 5d) that the short PRLR isoforms are expressed at relatively low levels. What is the relative expression in mouse splenocytes? This would be helpful in understanding any role for the short isoforms in their results.
2. Similarly, the functions of the human intermediate isoform of the PRLR are poorly understood. The recent report in the context of breast cancer (Gribble et al., *NJP Breast Cancer*, 2021) suggests that non-JAK2/Stat5 signals may be important in driving disease under some circumstances, and has renewed interest in this isoform. Is the intermediate PRLR isoform expressed in these cell lines? If so, it would be helpful to show the relative levels of long: intermediate PRLR to strengthen the argument for altered ratios of PRLR isoforms as an underlying mechanism.
3. In lines 112-114, the authors state that reducing expression of the "LFPRLR on B cells in vivo".... Note that their approach would reduce LFPRLR in vivo systemically in multiple tissues, although they focus on effects in the immune system here. This should be rephrased.
4. Recognizing that treatment was completed and analyses were carried out prior to development of DLBCL in most mice in that model, 4 months of age is none-the-less at an age when the authors state disease can be detected (line 217). It is of course of interest if reduction in LFPRLR can reduce disease risk. Was disease observed in any of these animals?

Minor:

1. It appears that the in vitro studies were all conducted in 10% FBS. If this is correct, then it is possible that the bovine prolactin in the media may also contribute to total prolactin agonist exposure (not just autocrine prolactin). Line 269: were these cells cultured in FBS?

Reviewer #2 (Remarks to the Author):

In the manuscript entitled "Isoform-specific knockdown of long and intermediate prolactin receptors interferes with evolution of B-cell neoplasms", Sawminathan and colleagues described novel findings that malignant human B cells express prolactin (PRL), which has the potential to engage the long isoform (LF) and intermediate form (IF) of the PRL receptor (LF/IFPRLR) to promote the proliferation and/or survival. This was addressed by using the knockdown approach in murine lupus-prone model (MRL mice) and B cell lymphomagenesis-prone model (TCL1 transgenic mice), as well as the several well-studied human DLBCL cell lines, with partially overlapping underlying mechanisms. The pro-neoplasm role of the PRL-LF/IFPRLR axis was not operational in normal B cells, which did not secrete PRL and expressed only the short form (SFPRRL). The findings are novel and have important clinical implications, with appropriate statistical analyses. Several minor points are as follows:

1. At least one Western Blot should be shown to confirm the reduced expression of LF/IFPRLR protein(s) after knockdown. This should be feasible in DLBCL cell line knockdown experiments in vitro (Figure 5I-J).
2. While MRL mice do develop severe lymphoproliferative symptoms, whether lupus carries increased risk of cancer is inconclusive (the cited references are more of anecdotal evidence). For this manuscript, it is not necessary to make this point anyways. The authors can simply test the hypothesis that the PRL-LF/IFPRLR axis plays an important role in B cell dysregulation, including that in lupus B cells and B lymphoma cells. Related to this issue, there is no need to emphasize stages of B cell malignancy – the three examples do not entirely fit the descriptions. If the authors really want to address the stage-specific role of LF/IFPRLR, much more sophisticated genetical modified mouse models (such as timed gene deletion) would be necessary.
3. Autoantibodies should be analyzed in MRL mice.
4. The sentence in Line 170-172 needs to be re-written to be clear.
5. The statement in Line 196-197 needs to be corrected, as no Annexin V-PI type experiments were performed to support the idea that the knockdown induces more B cell death in MRL mice. Again, the wording on the stage of lymphomagenesis should be avoided.
6. Figure 3C and Figure 4I are models and should be separated from the data, either as a main figure cited in the Discussion or as a supplementary figure.
7. In Figure 4d, no data point corresponds to the 60% of B cells in control group shown in the FACS plot.
8. ELISA should be performed to confirm the secretion of PRL by B lymphoma cell lines.
9. Overall, the manuscript could be re-organized to first show the relevance of PRL in lymphoma patients, then address the impact of knockdown in DLBCL cells in vitro, followed by in vivo studies in TCL1 transgenic mice and extended to a different model with shared B cell dysregulation (lupus).

Reviewer #3 (Remarks to the Author):

Prolactin and its receptors have been implicated in several types of cancer originating from reproductive or non-reproductive tissues. Moreover, previous studies have revealed that different isoforms of prolactin receptors play antagonistic roles in breast cancer metastasis. Built upon earlier work from some of the co-authors of this manuscript, Khani et al. used splice-modulating oligo (SMO) to selectively knock down the long and intermediate isoforms of prolactin receptors in the context of B-cell neoplasms and reported that targeting long and intermediate isoforms of prolactin receptors interfered with the evolution of B-cell malignancies.

The major strength of the paper is the use of three models to represent the initiation, establishment, and maintenance of B-cell malignancies, and thorough analysis of B cell phenotypes. However, the conclusion that targeting long and intermediate isoforms of prolactin receptors is of therapeutic value in preventing and treating B-cell malignancies is hampered by the lack of direct efficacy data. The complications of SLE-prone Mrl-lpr and DLBCL-prone TCL1-tg mouse models prevented the collection of efficacy data. However, the human DLBCL cell line models are amenable to CDX analyses. Demonstrating that SMO treatment can reduce disease burden in DLBCL CDX models will significantly strengthen the impact of the current study.

Minor points are listed below:

1. The authors solely relied on qPCR to evaluate isoform-selective modulation of prolactin receptor expression. Antibodies recognizing all isoforms of prolactin receptors are available. The authors should corroborate their conclusion using western blotting when enough protein lysates can be collected. In situations with limited materials, e.g., sorted B cells, the authors should perform RNA-seq and use genome browser tracks to show isoform-specific expression.
2. In Figure 3c, the author seems to suggest that B-cell transformation happens only in B cells in the spleen. Is this correct? Can B cell transformation occur in the bone marrow?
3. The role of prolactin in the context of B-cell malignancy is unclear. This can be evaluated by knocking out prolactin by CRISPR-Cas9 in human DLBCL cell lines and monitor cellular fitness over time.
4. In Figure 4h, FACS plots displayed two cell populations of different BCL2 staining intensities in the LFPRLR SMO group. Can the authors explain this phenomenon?

Response to Editor and Reviewers COMMSBIO-22-3083-T (Taghi Khani et al.)

We thank the Reviewers for their highly positive review of our study. The Reviewers' excellent suggestions have enabled us to submit a much-improved revised manuscript. We have addressed the comments by performing additional experiments (indicated as 'New Figures' in response) and/or revising the text. In the manuscript, new data and text are highlighted in grey.

Key New Figures in the manuscript are summarized below:

(1) We show the efficacy of LFPRLR knockdown in blocking the progression of overt human B-cell malignancies *in vivo* in two independent cell-line derived (CDX) models of DLBCL, which exhibited varying degrees of sensitivity to treatment with LFPRLR SMO *in vitro* (Summary Fig. 1, New Fig. 6 in manuscript).

Summary Fig. 1

(2) We now show the secretion of (Summary Fig. 2a, New Fig. 7a in manuscript) and requirement for (Summary Fig. 2b, New Fig. 7b in manuscript) autocrine PRL in the maintenance of malignant human B cells.

Summary Fig. 2

(3) The Reviewers requested immunoblots for PRLR isoforms to supplement the transcript expression data. Unfortunately, none of the commercially available antibodies could detect and measure the PRLR isoform proteins, which are expressed at low levels in all our models. Hence, as a surrogate for an immunoblot of the PRLR isoforms, we show a reduction in p-STAT3 after LFPRLR knockdown in malignant human B cells (Summary Fig. 3, New Fig. 5j in manuscript). These data confirm reduced expression of the LFPRLR protein since only the LFPRLR can bind STAT3 and activate it in response to PRL; only the LFPRLR has the appropriate intracellular sequence, derived from exon 10 during splicing of the PRLR pre-mRNA.

Summary Fig. 3

Response to Editor

Please note that the following revisions would be necessary for us to contact our referees again:

(i) Address if reduction of LFPRLR can decrease disease risk:

We have addressed this **in response to Reviewer 1: Q4**. We showed the efficacy of LFPRLR SMO in slowing progression of B-cell malignancies (disease risk) *in vivo* through CDX models (**Summary Fig. 1 above/ New Fig. 6 in manuscript; Manuscript Lines 325-338**).

As noted in **lines 236-254 and 408-418 in Manuscript**, while the DLBCL- and SLE-prone mouse models we used were not suitable for survival studies, *we measured indicators of disease risk specific to the age of the animals when we completed treatment with LFPRLR SMO*:

1. All **DLBCL-prone** mice hemizygous for the *TCL1* transgene develop signs of increased risk of DLBCL, including mild lymphocytosis and slight increases (<2-fold) in WBC counts beginning at 4 months of age. However, the majority (~95%) of mice become visibly ill with symptoms of overt DLBCL between 7-12 months of age (Hoyer et al., 2002, **Ref 28 in manuscript**). This made detection of overt disease in our treated animals (n=8-15 per group) highly improbable because all animals treated with SMOs were euthanized at 16 weeks of age (~4 months).

We measured WBC counts and lymphocytosis (indicators of disease risk in 4-month-old mice), and showed that LFPRLR knockdown reduced both (**New Figs. 4c-d**). Additionally, we show B-cell specific reductions in expressions of BCL2 (**Fig. 4f-g**) and TCL1 (**New Fig. 4e**) oncoproteins after LFPRLR knockdown. Hence, we conclude that LFPRLR knockdown reduces overt lymphoma risk by reducing pre-malignant B cell numbers and their expression of oncoproteins (**Manuscript Lines 236-254**).

2. **In SLE-prone mice**, reduction in disease risk after LFPRLR knockdown was shown by reductions in deleterious B cells (B and plasma cell subset numbers, aberrant B cells with long CDR3s), anti dsDNA antibodies (**New Fig. S4**), and BCL2 and AID expression in B cells.

(ii) Clarify the stage-specific role of LF/IFPRLR

We have addressed this **in response to Reviewer 2: Q2**. To delineate LFPRLR's role in the evolution of B-cell malignancies, we interrogated models with dysregulated B cells in increasing order of their malignancy status: non-malignant but aberrant B cells that have increased risk of malignancy initiation (e.g., SLE), pre-malignant B cells with constitutive activation of an oncogene yet to become overt disease, and overt B-cell malignancies. *We realized that we had incorrectly used the word 'stage' to depict this increasing order of deleteriousness of B cells and thank the Reviewer for pointing this out. Following the Reviewer's important suggestion, we have now deleted the word 'stage' throughout the manuscript.*

(iii) Include the experiment demonstrating that SMO treatment can reduce disease burden in DLBCL CDX models to strengthen the impact of the current study.

In response to Reviewer 3, we now show that LFPRLR knockdown reduces disease burden in two independent models of human DLBCL CDX (**Summary Fig. 1, New Fig. 6 in manuscript; Manuscript Lines 325-338**).

Response to Reviewers

Reviewer #1

Although early studies of prolactin on the immune system did not reveal strong activity in health, its actions are increasingly recognized in the context of underlying pathology. Here the authors investigate prolactin signals in B cell malignancies, employing mouse models, and human data sets, samples and cell lines. They utilize an innovative approach to reduce canonical prolactin signals by modulating splicing of prolactin receptor transcripts, and investigate the consequences in *in vivo* mouse models of systemic lupus erythematosus (SLE) and B cell lymphoma. A relatively small change in ratio of LFPRLR/ total PRLR mRNAs in splenic leucocytes reduces the number of B but not T cells and plasmacytoid dendritic cells in the SLE model, associated with reduced cell cycle progression and reduced Bcl2 and Aid mRNA in splenic B cells, both recognized players in clinical disease. In the DLBCL model, this treatment reduces B cells in the spleen, although it increases them in the bone marrow. They explore a role in human clinical disease by interrogating publicly available data sets of patient DLBCL and ALL tumors, identifying an association of high expression of prolactin with a worse prognosis. Notably, prolactin mRNA is very low in B cells of healthy patients, but is much higher in patient tumors and several B cell tumor cell lines. *In vitro* studies in multiple human B cell tumor lines indicate that prolactin plays a role in cell lines with BCL2 translocations, but not both MYC/BCL2 translocations. These studies in multiple models underscore the importance of canonical prolactin signals in various states of B cell pathology, and emphasize the changing expression of components of this pathway in disease. Their studies identify specific sites of action of prolactin, both physical location and relevant molecular targets. Together, their findings illuminate a role for prolactin in both increasing the risk for B cell pathologies and in driving a subset of DLBCL tumors, and suggest characteristics of patient tumors that may respond to directed therapies.

We thank the Reviewer for their highly positive review of our manuscript and detail below how we have answered individual questions.

Addressing the following would be helpful:

1. It is interesting that very substantial effects on biology are initiated by a relatively small, although significant, reduction in the ratio of long form PRLR mRNA, compared to total PRLR mRNA. It is shown in human cells (Fig. 5d) that the short PRLR isoforms are expressed at relatively low levels. What is the relative expression in mouse splenocytes? This would be helpful in understanding any role for the short isoforms in their results.

We thank the Reviewer for their question. Of the mouse PRLR isoforms, LF, SF1, SF2, and SF3 (Ormandy, C, J, **Ref 22 in manuscript**), only SF3 and LF were found to be expressed in splenic leukocytes of the mouse strains we used. As in human B cells, expression of the SFPRLR is several-fold lower than the LFPRLR in mouse splenocytes. **Figure below** shows the absolute expressions of PRLR isoforms in splenic leukocytes in our mouse models: each stacked bar represents a single mouse.

As the Reviewer will appreciate from the figure above, there is a huge degree of variability in the absolute expression of the isoforms across individual mice in each strain. To account for this variability and to most accurately predict changes in pro-proliferative, anti-apoptotic effects of PRLRs in cells after LFPRLR knockdown, expression of PRLR isoforms in each mouse was represented as the ratio of LFPRLR: total PRLR. An exception to calculating the ratio is made in instances where the cells express only LF/IFPRLR, as in some malignant human B cells in **Figs.5d, 7c, 7d, S13**. In the **Methods** section (**Manuscript Lines 468-474**), we explain why we compare the ratio of LFPRLR: total PRLR in **Figs. 1b, 2a, 4b, S10**.

2. Similarly, the functions of the human intermediate isoform of the PRLR are poorly understood. The recent report in the context of breast cancer (Grible et al., NJP Breast Cancer, 2021) suggests that non-JAK2/Stat5 signals may be important in driving disease under some circumstances and has renewed interest in this isoform. Is the intermediate PRLR isoform expressed in these cell lines? If so, it would be helpful to show the relative levels of long: intermediate PRLR to strengthen the argument for altered ratios of PRLR isoforms as an underlying mechanism.

We appreciate the Reviewer's question and have tried to address this with the following new experiments:

1. Our previous qPCR primers for the human cell lines specifically detected the LFPRLR. As the Reviewer suggested, we designed new primers to specifically detect the IF. All human malignant B-cell lines used in this study express 10-80-fold more LF than IF, and, as expected, the LFPRLR SMO knocked down expression of both the LF and IF (**New Fig. S13**). We cannot exclude a role for LF-IF heterodimers or IF-IF homodimers in growth and viability of these cell lines. However, the effect of LFPRLR SMO on viability (**Fig 5i**) inversely correlates with the ratio of LF:IF, thereby suggesting that LF plays a predominant role (**see Figure below**).

2. The complete exon 10 of the LFPRLR binds STATs (e.g., STAT3, 5) and others have shown the importance of activated STAT3 to Bcl2-mediated B cell survival (Flores-Fernández, R. et al., **Ref 18 in manuscript**). As the Reviewer points out, SF and IF PRLRs cannot bind and activate STATs, although they can bind the JAKs (see **Manuscript Lines: 100-111**). Consistent with the high expression of LF in comparison to the IFPRLR in B-cell lymphomas (**New Fig. S13**), we observed that treatment with LFPRLR SMO reduced the phosphorylation (activation) and production of STAT3 in three out of four malignant B cell lines (**New Fig. 5j**). PSTAT5 was not detected in these cells, even in the absence of LFPRLR knockdown, although a reduction in global STAT5 was observed at some time points after LFPRLR knockdown (**New Fig. S15**). Thus, the Jak-STAT3 signaling pathway may play an important role downstream of the LFPRLR in human B-cell lymphomas. However, the Karpas-422 line, which is very sensitive to the LFPRLR SMO, creates some level of doubt since the LFPRLR SMO did not affect either STAT3 or STAT5. Future studies will further investigate mechanism of action of LFPRLR SMO in human DLBCL cells since it is either not straightforward or varies with cell line.

3. In lines 112-114, the authors state that reducing expression of the “LFPRLR on B cells in vivo”. Note that their approach would reduce LFPRLR in vivo systemically in multiple tissues, although they focus on effects in the immune system here. This should be rephrased.

We have now reworded as “To test this, we measured the effects of systemically knocking down expression of the LFPRLR on immune cells: (1) *in vivo* in mouse models prone to SLE carrying non-malignant but aberrant B cells, (2) *in vivo* in mouse models prone to DLBCL having pre-malignant B cells, and (3) *in vitro* and *in vivo* in overt malignant human B-cell lines along with knockdown of IFPRLR.” (see **Manuscript Lines 115-119**).

4. Recognizing that treatment was completed and analyses were carried out prior to development of DLBCL in most mice in that model, 4 months of age is none-the-less at an age when the authors state disease can be detected (line 217). It is of course of interest if reduction in LFPRLR can reduce disease risk. Was disease observed in any of these animals?

We thank the Reviewer for this important question. According to Hoyer et al., who developed the model, <5% of the mice become “visibly ill” at ~4 months of age and no changes in lymphoid organ structure or in early and late B-lymphopoiesis are detected in most 4-month-old mice. Most mice die from overt DLBCL between 7-12 months of age. This made detection of overt disease in our treated animals (n=8-15 per group) highly improbable because all animals were euthanized at 16 weeks (~4 months) when we completed treatment with SMOs.

Hoyer et al. show that the major indicators of disease risk in 4-month-old *TCL1-tg* DCBL-prone mice include mild lymphocytosis and slight increases (<2-fold) in WBC counts. Therefore, we used WBC and B-cell counts as **indicators for disease risk** and show that LFPRLR knockdown reduces both (**New Figs. 4c, 4d**). We also show reduced expression of the *BCL2* (**Fig. 4f**), and *TCL1* (**New Fig. 4e**) oncoproteins in B cells after LFPRLR knockdown in DLBCL-prone mice as **indicators of reduced lymphoma risk** because high levels of these proteins are known to predict poor clinical prognosis in patients with DLBCL (Hu, S. et al. 2013; Aggarwal, M. et al. 2009; Ramuz, O. et al. 2005, **Ref 65-67 in manuscript**). (**Manuscript Lines 236-254, 384-386**).

Finally, as Reviewer 3 acknowledged, “The complications of SLE-prone *Mrl-lpr* and DLBCL-prone *TCLI-tg* mouse models prevented the collection of efficacy data.” Hence, we followed their suggestion and showed the efficacy of LFPRLR SMO in slowing progression of B-cell malignancies (disease risk) *in vivo* through CDX models (**Summary Fig. 1 in Response: Page 1/ New Fig. 6 in manuscript**). We have also acknowledged our inability to measure how LFPRLR drives the establishment of ‘actual overt lymphomas’ as a limitation in the Discussion (**Manuscript Lines 325-338**).

Minor:

1. It appears that the *in vitro* studies were all conducted in 10% FBS. If this is correct, then it is possible that the bovine prolactin in the media may also contribute to total prolactin agonist exposure (not just autocrine prolactin). Line 269: were these cells cultured in FBS?

The Reviewer raises a valid concern. However, it has been shown by others that PRLRs on human cells do not respond to non-human PRLs (Utama et al., *Endocrinology*, 2009, **Ref 61 in manuscript**). Therefore, bovine PRL from FBS-supplemented media should not influence the human lymphoma cell lines. Additionally, in response to Reviewers 2 and 3, we have also demonstrated the secretion and requirement of autocrine human PRL in maintenance of human B-cell lymphoma lines (**New Figs. 7a-7b**).

Reviewer #2

In the manuscript entitled “Isoform-specific knockdown of long and intermediate prolactin receptors interferes with evolution of B-cell neoplasms”, Swaminathan and colleagues described novel findings that malignant human B cells express prolactin (PRL), which has the potential to engage the long isoform (LF) and intermediate form (IF) of the PRL receptor (LF/IFPRLR) to promote the proliferation and/or survival. This was addressed by using the knockdown approach in murine lupus-prone model (MRL mice) and B cell lymphomagenesis-prone model (TCL1 transgenic mice), as well as the several well-studied human DLBCL cell lines, with partially overlapping underlying mechanisms. The pro-neoplasm role of the PRL-LF/IFPRLR axis was not operational in normal B cells, which did not secrete PRL and expressed only the short form (SFPRLR). The findings are novel and have important clinical implications, with appropriate statistical analyses.

We thank the Reviewer for their highly positive review of our manuscript and detail below our responses to each individual comment.

Several minor points are as follows:

1. At least one Western Blot should be shown to confirm the reduced expression of LF/IFPRLR protein(s) after knockdown. This should be feasible in DLBCL cell line knockdown experiments *in vitro* (Figure 5I-J).

As the Reviewer suggests in their comment, the low expression of PRLR isoforms in B cells and the inability to obtain sufficient sorted B cells precluded us from conducting immunoblots for PRLR isoforms in mouse B cells. Following the Reviewer’s suggestion, we tried to measure PRLR isoforms by western blot in human DLBCL cells. Unfortunately, none of the commercially available anti-PRLRs (including the one used in Grible et al., 2021, **Ref 23 in manuscript**) could detect the endogenous expression of PRLR isoform proteins, even in human DLBCL cell lines. We politely point out that others (e.g., Grible et al., 2021), only show PRLR isoform proteins by immunoblotting after overexpressing these isoforms in CHO cells and not the endogenous levels of these isoforms in cancer cells. In addition, although some commercially available antibodies produce bands of approximately the correct molecular weights, these bands were present in negative control cells (e.g. HEK) indicating they are non-specific. Sufficiently sensitive anti-PRLRs that can accurately detect low endogenous levels of the PRLR isoforms remain to be developed.

As a surrogate for specific modulation of the ratio of LF: total PRLR proteins by the LFPRLR SMO, we show ablation of the LFPRLR-induced STAT3 activation after LFPRLR knockdown. LFPRLR SMO prevents the inclusion of exon 10 of LFPRLR that has the potential to bind STAT proteins (e.g., STAT3, 5) in order that they can be phosphorylated by Jak. SFPRLRs cannot bind STATs. Therefore, we posited that STAT activation should be a reliable indicator of the relative levels of PRLR isoforms. LFPRLR knockdown reduced the phosphorylation (activation) and production of STAT3 in three out of four malignant B-cell lines used in our study (New Fig 5j).

We also measured the p-STAT3 levels in 4 samples each from control SMO- and LFPRLR SMO-treated SLE-prone and DLBCL-prone mice. Using a p value of 0.05 as the cutoff, we saw a non-

statistically significant trend toward reduction in pSTAT3 after LFPRLR knockdown in B cells in both models (figure shown below).

2. While MRL mice do develop severe lymphoproliferative symptoms, whether lupus carries increased risk of cancer is inconclusive (the cited references are more of anecdotal evidence). For this manuscript, it is not necessary to make this point anyways.

We politely disagree on this point since SLE has been shown to be associated with increased risk of B-cell malignancies in a sizeable cohort of patients. Nevertheless, we have deleted all references that provide anecdotal evidence in the form of case reports and only retained references that provide evidence of lymphoma development in a sizeable cohort of patients with SLE (Bernatsky, S. et al. 2013; Bernatsky, S. et al., 2014; Koff, J. L. & Flowers, C. R. 2016, **Ref 3-5 in manuscript**). In mouse models, B-cell transformation has been specifically observed in the SLE-prone *MRL-lpr* model (White, C. A. et al. 2011, **Ref 6 in manuscript**), which makes this a good model to interrogate the role of LFPRLR in increasing the risk of B-cell malignancy development.

The authors can simply test the hypothesis that the PRL-LF/IFPRLR axis plays an important role in B cell dysregulation, including that in lupus B cells and B lymphoma cells. Related to this issue, there is no need to emphasize stages of B cell malignancy – the three examples do not entirely fit the descriptions.

Because the goal of our study was to delineate LFPRLR's role in the evolution of B-cell malignancies, we interrogated models with dysregulated B cells in increasing order of their malignancy status: non-malignant but aberrant B cells that have increased risk of malignancy initiation (e.g., *MRL-lpr* SLE-prone mice), pre-malignant B cells with constitutive activation of an oncogene yet to become overt disease, and overt B-cell malignancies. *We realized that we had incorrectly used the word 'stage' to depict this increasing order of deleteriousness of B-cells and thank the Reviewer for pointing this out. We have now deleted the word 'stage' throughout the manuscript.*

If the authors really want to address the stage-specific role of LF/IFPRLR, much more sophisticated genetically modified mouse models (such as timed gene deletion) would be necessary.

The Reviewer rightly notes that we did not interrogate the stage-specific role of LFPRLR in B cells and that clarifying the stage-specific role of LFPRLR would require timed deletion of

LFPRRLR during specific stages of B-cell development. This is an excellent suggestion, one we propose to interrogate in future studies.

3. Autoantibodies should be analyzed in MRL mice.

We appreciate the Reviewer's suggestion, even though our intent in this study was not to examine SLE per se. As mentioned above, the *Mrl-lpr* mouse was used as a model in which non-malignant but aberrant B cells that have an increased risk of transformation accumulate (White, C.A. et al., 2011, **Ref 6 in manuscript**). Nevertheless, in light of a recent study showing that PRL drives the production of anti-dsDNA autoantibodies in *MRL-lpr* mice (Carreon-Talavera et al, 2022, **Ref 43 in manuscript**), we have examined the effect of LFPRRLR SMO on anti-dsDNA autoantibody production, mostly because Carreon-Talavera et al. did not determine which form of the PRLR was important for such production. Within the time frame of treatment, we observed a trend towards reduction in autoantibody levels after LFPRRLR knockdown (**New Fig. S4**). Because of the reduction in all plasma cell subsets and reduction in frequencies of potentially autoreactive, aberrant B cells with long CDR3s>20aa after LFPRRLR knockdown (**Figs. 1f and 2e**), we predict longer term treatment with the LFPRRLR SMO to further reduce production of these and other autoantibodies, including anti-histones and anti-cardiolipins.

4. The sentence in Line 170-172 needs to be re-written to be clear.

We have rewritten this sentence to explain how the lack of changes in BCL2 protein after LFPRRLR knockdown in splenic B cells is consistent with a reduction in B-cell proliferation. We have rephrased as “The anti-apoptotic protein BCL2 is known to promote B-cell survival but not proliferation (McDonnell, T. J. et al. 1990, **Ref 44 in manuscript**). Our findings showing unchanged expression of BCL2 protein and reduction in B-cell proliferation after LFPRRLR knockdown are consistent with this.” (see **Manuscript Lines 189-192**)

5. The statement in Line 196-197 needs to be corrected, as no Annexin V-PI type experiments were performed to support the idea that the knockdown induces more B cell death in MRL mice. Again, the wording on the stage of lymphomagenesis should be avoided.

We thank the Reviewer for pointing this out. As suggested, we have now rephrased this sentence as “Thus, the PRL-LFPRRLR axis promotes the retention of deleterious B cells, thereby increasing the risk of lymphoma development.” (see **Manuscript Lines 215-217**)

6. Figure 3C and Figure 4I are models and should be separated from the data, either as a main figure cited in the Discussion or as a supplementary figure.

We respect the Reviewer's suggestion. However, we felt retention of these graphics, describing the LFPRRLR-mediated B cell regulation in each model of B-cell dysregulation would aid in better understanding, and have therefore retained them in their original locations.

7. In Figure 4d, no data point corresponds to the 60% of B cells in control group shown in the FACS plot.

We politely point out to the Reviewer that while the FACS plot shows percentages of B cells, the quantitation in **Fig. 4d** shows B-cell numbers. B cell numbers that correspond to the representative FACS plots are control SMO = 33.8×10^6 , LFPRRLR SMO = 26.6×10^6)

8. ELISA should be performed to confirm the secretion of PRL by B lymphoma cell lines.

The Reviewer makes an excellent suggestion. We have now confirmed the secretion of PRL by B-lymphoma cell lines. Of note, the cell lines sensitive to nM concentrations of LFPRLR SMO *in vitro* (Karpas-422 and SU-DHL-6) secrete significantly more PRL than their counterparts that are sensitive to μ M concentrations of LFPRLR SMO (VAL and OCI-LY-18) (**New Fig. 7a**). Along these lines, we also show that neutralization of secreted PRL markedly affects the viability of especially those cell lines that secrete high amounts of PRL (Karpas-422 and SU-DHL-6). These findings demonstrate the requirement for autocrine human PRL in the maintenance of human B-cell lymphoma lines (**New Figs. 7a-b**) (see **Manuscript Lines 345-354**).

9. Overall, the manuscript could be re-organized to first show the relevance of PRL in lymphoma patients, then address the impact of knockdown in DLBCL cells *in vitro*, followed by *in vivo* studies in TCL1 transgenic mice and extended to a different model with shared B cell dysregulation (lupus).

While we appreciate this suggestion, we have retained the current flow of the manuscript because to describe the role of LFPRLR in B-cell malignancy evolution, we felt it would be better to describe LFPRLR's role in dysregulated B cells in increasing order of their malignancy status: non-malignant but aberrant B cells that have increased risk of malignancy initiation (e.g., *MRL-lpr* SLE-prone mice), pre-malignant B cells with constitutive activation of an oncogene yet to become overt disease, and overt B-cell malignancies.

Reviewer #3

Prolactin and its receptors have been implicated in several types of cancer originating from reproductive or non-reproductive tissues. Moreover, previous studies have revealed that different isoforms of prolactin receptors play antagonistic roles in breast cancer metastasis. Built upon earlier work from some of the co-authors of this manuscript, Khani et al. used splice-modulating oligo (SMO) to selectively knock down the long and intermediate isoforms of prolactin receptors in the context of B-cell neoplasms and reported that targeting long and intermediate isoforms of prolactin receptors interfered with the evolution of B-cell malignancies. The major strength of the paper is the use of three models to represent the initiation, establishment, and maintenance of B-cell malignancies, and thorough analysis of B cell phenotypes. However, the conclusion that targeting long and intermediate isoforms of prolactin receptors is of therapeutic value in preventing and treating B-cell malignancies is hampered by the lack of direct efficacy data. The complications of SLE-prone *Mrl-lpr* and DLBCL-prone *TCL1-tg* mouse models prevented the collection of efficacy data. However, the human DLBCL cell line models are amenable to CDX analyses. Demonstrating that SMO treatment can reduce disease burden in DLBCL CDX models will significantly strengthen the impact of the current study.

We thank the Reviewer for the positive review of our manuscript. As Reviewer 3 acknowledged, “The complications of SLE-prone *Mrl-lpr* and DLBCL-prone *TCL1-tg* mouse models prevented the collection of efficacy data.” As discussed above, we followed the excellent suggestion to show the efficacy of LFPRLR SMO in slowing progression of B-cell malignancies (disease risk) *in vivo* through CDX models (**Summary Fig. 1 in Response: Page 1/ New Fig. 6 in manuscript; Manuscript Lines 325-338**).

Minor points are listed below:

1. The authors solely relied on qPCR to evaluate isoform-selective modulation of prolactin receptor expression. Antibodies recognizing all isoforms of prolactin receptors are available. The authors should corroborate their conclusion using western blotting when enough protein lysates can be collected.

As the Reviewer rightly points out, the low expression of PRLR isoforms in B cells and the inability to obtain sufficient sorted B cells precluded us from conducting immunoblots for PRLR isoforms in mouse B cells. Following the Reviewer’s suggestion, we tried to measure PRLR isoforms by western blot in human DLBCL cells. Unfortunately, none of the commercially available anti-PRLRs (including the one used in Gribble et al., 2021, **Ref 23 in manuscript**) could detect the endogenous expression of PRLR isoform proteins, even in human DLBCL cell lines. We politely point out that others (e.g., Gribble et al., 2021), only show PRLR isoform proteins by immunoblotting after overexpressing these isoforms in CHO cells and *not the endogenous levels of these isoforms in cancer cells*. In addition, although some commercially available antibodies produce bands of approximately the correct molecular weights, these bands were present in negative control cells (e.g. HEK) indicating they are non-specific. Sufficiently sensitive anti-PRLRs that can accurately detect low endogenous levels of the PRLR isoforms remain to be developed.

As a surrogate for specific modulation of the ratio of LF: total PRLR proteins by the LFPRLR SMO, we show ablation of the LFPRLR-induced STAT3 activation after LFPRLR knockdown. LFPRLR SMO prevents the inclusion of exon 10 of LFPRLR that has the potential to bind STAT proteins (e.g., STAT3, 5) in order that they can be phosphorylated by Jak. SFPRLRs cannot bind STATs. Therefore, we posited that STAT activation should be a reliable indicator of the relative levels of PRLR isoforms. LFPRLR knockdown reduced the phosphorylation (activation) and production of STAT3 in three out of four malignant B-cell lines used in our study (New Fig. 5j).

A reduction in global STAT5 was observed after LFPRLR knockdown, although pSTAT5 was undetected in these cells even in the absence of LFPRLR knockdown (New Figs S15). Hence, LFPRLR-mediated B-cell lymphoma growth and survival are driven predominantly by STAT3. The above findings confirm the mechanism of action of the LFPRLR SMO and at least partly circumvent the technical difficulty of measuring changes in protein levels of PRLR isoforms.

We also measured the p-STAT3 levels in 4 samples each from control SMO- and LFPRLR SMO-treated SLE-prone and DLBCL-prone mice. Using a p value of 0.05 as the cutoff, we saw a non-statistically significant trend toward reduction in pSTAT3 after LFPRLR knockdown in B cells in both models (figure shown below).

In situations with limited materials, e.g., sorted B cells, the authors should perform RNA-seq and use genome browser tracks to show isoform-specific expression.

We politely point to the Reviewer that sorted B cells from the mouse models were not sufficient for conducting bulk RNA-seq. Furthermore, RNA-seq would require qPCR validation. Therefore, when possible, we directly used the sorted B cells to quantify the expression of PRLR isoforms by qPCR (e.g., SLE-prone mice in Fig. 2a). When sufficient B cells could not be sorted even for qPCR, we measured PRLR isoforms in bulk splenic leukocytes by qPCR (e.g., DLBCL-prone mice in Fig. 4b). When sorting does not yield sufficient cells for qPCR, conducting 3' single cell RNA sequencing (scRNAseq) on unsorted samples would be the best method to characterize the B-cell specific expression of PRLR isoforms. Unfortunately, without additional funds, we cannot conduct scRNA-seq experiments.

2. In Figure 3c, the author seems to suggest that B-cell transformation happens only in B cells in the spleen. Is this correct? Can B cell transformation occur in the bone marrow?

The Reviewer raises an important question. Signaling through the LFPRLR increases the risk of B-cell transformation in splenic B cells through induction of AID expression which is known to drive B-cell transformation (White, C.A. et al., 2011; Swaminathan et al., 2015; Tsai et al., 2008, **Refs 6, 9 and 49 in manuscript**). In the bone marrow, we checked AID expression in B cells and did not find changes after LFPRLR knockdown (shown below), indicating that transformation may not be directly impacted. However, the reduction in the BCL2 oncoprotein in bone marrow B cells after LFPRLR knockdown suggests that LFPRLR increases B-cell populations available for transformation and malignancy initiation in the bone marrow. We have now corrected Figure 3c. to reflect this change (**New Fig. 3c**).

3. The role of prolactin in the context of B-cell malignancy is unclear. This can be evaluated by knocking out prolactin by CRISPR-Cas9 in human DLBCL cell lines and monitor cellular fitness over time.

The Reviewer makes an excellent suggestion which was also made by Reviewer 2. As suggested by Reviewer 2 in Q8, we first measured the secretion of PRL from DLBCL cell lines using ELISA. Of note, the cell lines sensitive to nM concentrations of LFPRLR SMO (Karpas-422 and SU-DHL-6) secrete significantly more PRL than their counterparts that are only sensitive to higher (μ M) concentrations of LFPRLR SMO (VAL and OCI-LY-18) (**New Fig. 7a**). Then, as suggested by Reviewer 3, we also now show that neutralization of secreted PRL markedly affects viability of especially those cell lines that secrete high amounts of PRL (Karpas-422 and SU-DHL-6). These findings demonstrate the requirement for autocrine human PRL in the maintenance of human B-cell lymphoma lines (**New Figs. 7a-b**) (see **Manuscript Lines 345-354**). We would like to point out to Reviewer 3 that we tried but could not successfully maintain human DLBCL cells transduced with Cas9, and hence decided to address this question by neutralizing the secreted PRL and measuring cell fitness.

4. In Figure 4h, FACS plots displayed two cell populations of different BCL2 staining intensities in the LFPRLR SMO group. Can the authors explain this phenomenon?

The Reviewer correctly notes that there are two peaks with different staining intensities (MFI) of BCL2 in the LFPRLR SMO-treated group which are distinct from the fluorescence minus one (FMO) negative control. Additional analyses of our data indicate that LFPRLR knockdown increases the frequencies of B cells which are low expressers of BCL2 and concomitantly reduces the proportion of cells that are high expressers of BCL2 (**New Fig. 4i middle panel**).

REVIEWERS' COMMENTS:

Reviewer #1 (Remarks to the Author):

This manuscript addresses an important area of prolactin action, and the authors have addressed the majority of the reviewers' concerns.

Reviewer #2 (Remarks to the Author):

The authors have tried to address all the minor points raised by this reviewer. They have also completed additional experiments to address many concerns raised by the other reviewers. As such, the manuscript is of much better quality.

Reviewer #3 (Remarks to the Author):

The authors have sufficiently addressed all my comments. I recommend acceptance for publication.